# Sequential Algorithms for Testing Closeness of Distributions

**Omar Fawzi**
Univ Lyon, ENS Lyon, UCBL
CNRS, Inria, LIP, F-69342
Lyon Cedex 07, France
omar.fawzi@ens-lyon.fr

**Nicolas Flammarion**
EPFL
Lausanne, Switzerland
nicolas.flammarion@epfl.ch

**Aurélien Garivier**
UMPA UMR 5669 and LIP UMR 5668 CNRS,
ENS de Lyon, UCB Lyon 1
Lyon, France
aurelien.garivier@ens-lyon.fr

**Aadil Oufkir**
LIP UMR 5668 CNRS
ENS de Lyon, UCB Lyon 1
Lyon, France
aadil.oufkir@ens-lyon.fr

## Abstract

What advantage do *sequential* procedures provide over batch algorithms for testing properties of unknown distributions? Focusing on the problem of testing whether two distributions $\mathcal{D}_1$ and $\mathcal{D}_2$ on $\{1, \dots, n\}$ are equal or $\varepsilon$-far, we give several answers to this question. We show that for a small alphabet size $n$, there is a sequential algorithm that outperforms any batch algorithm by a factor of at least 4 in terms sample complexity. For a general alphabet size $n$, we give a sequential algorithm that uses no more samples than its batch counterpart, and possibly fewer if the actual distance $\mathrm{TV}(\mathcal{D}_1, \mathcal{D}_2)$ between $\mathcal{D}_1$ and $\mathcal{D}_2$ is larger than $\varepsilon$. As a corollary, letting $\varepsilon$ go to 0, we obtain a sequential algorithm for testing closeness when no a priori bound on $\mathrm{TV}(\mathcal{D}_1, \mathcal{D}_2)$ is given that has a sample complexity $\tilde{\mathcal{O}}(\frac{n^{2/3}}{\mathrm{TV}(\mathcal{D}_1, \mathcal{D}_2)^{4/3}})$: this improves over the $\tilde{\mathcal{O}}(\frac{n/\log n}{\mathrm{TV}(\mathcal{D}_1, \mathcal{D}_2)^2})$ tester of Daskalakis and Kawase [2017] and is optimal up to multiplicative constants. We also establish limitations of sequential algorithms for the problem of testing closeness: they can improve the worst case number of samples by at most a constant factor.

## 1 Introduction

How to test if two discrete sources of randomness are similar or distinct? This basic and ubiquitous question is surprisingly not closed if frugality matters, that is if one wants to take the right decision using as few samples as possible. To state the problem more precisely, one first needs to define what "distinct" means. In this paper, we endow the set of probability distributions on $\{1, \dots, n\}$ with the *total variation distance* TV, and we fix a tolerance parameter $\varepsilon \in [0, 1]$. We consider two distributions $\mathcal{D}_1$ and $\mathcal{D}_2$, and we assume that either $\mathcal{D}_1 = \mathcal{D}_2$ or $\mathrm{TV}(\mathcal{D}_1, \mathcal{D}_2) > \varepsilon$. Whenever $0 < \mathrm{TV}(\mathcal{D}_1, \mathcal{D}_2) \leq \varepsilon$, we do not expect any determined behaviour from our test. When both distributions are unknown we are *testing closeness*, based on an equal number of independent samples of both distributions.

35th Conference on Neural Information Processing Systems (NeurIPS 2021).

| Model | Lower bound | Upper bound |
|---|---|---|
| Batch | $4\log(1/\delta)\varepsilon^{-2} - \mathcal{O}(\log\log(1/\delta)\varepsilon^{-2})$ | $4\log(1/\delta)\varepsilon^{-2} + \mathcal{O}(n\varepsilon^{-2})$ |
| Sequential ($\tau_1$) | $\log(1/\delta)\varepsilon^{-2} - \mathcal{O}(\varepsilon^{-2})$ | $\log(1/\delta)\varepsilon^{-2}$ $+\mathcal{O}((n+\log(1/\delta)^{2/3})\varepsilon^{-2})$ |
| Sequential ($\tau_2$) | $\log(1/\delta)d^{-2} - \mathcal{O}(d^{-2})$ | $\log(1/\delta)d^{-2}$ $+\mathcal{O}\left((n+\log(1/\delta)^{2/3})d^{-2}\right)$ |

Table 1: Lower and upper bounds on the sample complexities for testing closeness in the batch and sequential settings with $d = \mathrm{TV}(\mathcal{D}_1, \mathcal{D}_2)$. The $\mathcal{O}$ hides universal constants.

We also need to specify what kind of "test" is considered. Here we treat the two hypotheses symmetrically (there is no "null hypothesis") : given a fixed risk $\delta \in (0, 1)$, we expect our procedure to find the true one with probability $1 - \delta$, whichever it is. We call such a procedure $\delta$-*correct*. Finally, we consider and compare two notions of "frugality": in the *batch* setting, the agent specifies in advance the number of samples needed for the test: she takes her decision just after observing the data all at once, and the sample complexity of the test is the smallest sample size of a $\delta-$correct procedure. In the *sequential* setting, the agent observes the samples one by one, and decides accordingly whether she takes her decision or requests to see more samples before making a decision. Then, the sample complexity of the test is the smallest *expected number of samples* needed before a $\delta$-correct procedure takes a decision. Note that this expected number can depend on the unknown distributions $\mathcal{D}_1$ and $\mathcal{D}_2$, which can lead to important advantages of sequential procedures.

**Contributions** When $n \geq 2$ is small, we show that the optimal sample complexities can be precisely characterized (up to lower order terms in $\delta$) in both the batch and sequential setting as shown in Table 1. This establishes a provable advantage for sequential strategies over batch strategies when $n \ll \log(1/\delta)$: sequential algorithms reduce the sample complexity by a factor of at least 4 for any pair of distributions $\mathcal{D}_1$ and $\mathcal{D}_2$. In addition, the sequential algorithms stop even more rapidly if the tested distributions are far (i.e., $\mathrm{TV}(\mathcal{D}_1, \mathcal{D}_2) \gg \varepsilon$). The improvements of the sequential algorithm are illustrated in Fig. 1. The sequential algorithms use stopping rules inspired from time uniform concentration inequalities. The problem of testing closeness for small $n$ is studied in Sec. 3.

For general $n \geq 2$, we improve the dependence on $\varepsilon$ to $\varepsilon \vee \mathrm{TV}(\mathcal{D}_1, \mathcal{D}_2)$ in the best batch algorithm due to Diakonikolas et al. [2020], which is known to be optimal up to multiplicative constants. Namely we obtain a sequential closeness testing algorithm using a number of samples given by

$$\mathcal{O}\left(\max\left(\frac{n^{2/3}\log^{1/3}(1/\delta)}{(\varepsilon \vee \mathrm{TV}(\mathcal{D}_1, \mathcal{D}_2))^{4/3}}, \frac{n^{1/2}\log^{1/2}(1/\delta)}{(\varepsilon \vee \mathrm{TV}(\mathcal{D}_1, \mathcal{D}_2))^2}, \frac{\log(1/\delta)}{(\varepsilon \vee \mathrm{TV}(\mathcal{D}_1, \mathcal{D}_2))^2}\right)\right). \tag{1}$$

As a special case, when $\varepsilon = 0$ (the algorithm should not stop when $\mathcal{D}_1 = \mathcal{D}_2$ in this case) we show that there is an algorithm that stops after

$$\mathcal{O}\left(\max\left(\frac{\log\log(1/d)}{d^2}, \frac{n^{2/3}\log\log(1/d)^{1/3}}{d^{4/3}}, \frac{n^{1/2}\log\log(1/d)^{1/2}}{d^2}\right)\right) \tag{2}$$

samples where $d = \mathrm{TV}(\mathcal{D}_1, \mathcal{D}_2) > 0$. This is an improvement over the sequential algorithm of Daskalakis and Kawase [2017] which uses $\Theta(\frac{n/\log n}{d^2}\log\log(1/d))$ samples. We design the stopping rules according to a time uniform concentration inequality deduced from McDiarmid's inequality, where we use the ideas of Howard et al. [2018, 2020] in order to obtain powers of $\log\log(1/d)$ instead of $\log(1/d)$.

We show that the sample complexity for the testing closeness problem given by Eq. (1) is optimal up to multiplicative constants in the worst case setting (i.e., when looking for a bound independent of the distributions $\mathcal{D}_1$ and $\mathcal{D}_2$). To do so, we construct two families of distributions whose cross TV distance is exactly $d \geq \varepsilon$ and hard to distinguish unless we have a number of samples given by Eq. (1). This latter lower bound is based on properties of KL divergence along with Wald's Lemma. We establish a lower bound on the number of queries that matches Eq. (2) up to multiplicative constants.

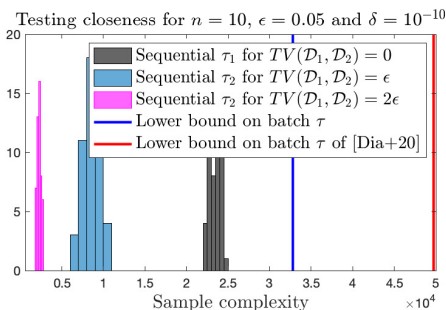
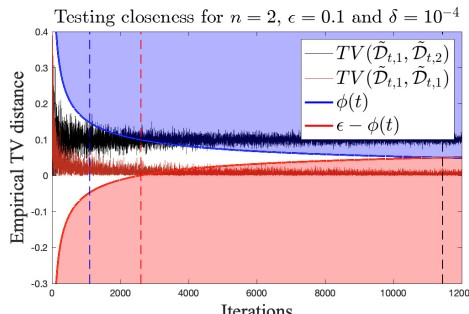

Figure 1: Left: histogram of the stopping times for 100 Monte-Carlo experiments. Black: $\mathcal{D}_1 = \mathcal{D}_2 = U_n$, blue (resp. magenta): $\mathcal{D}_1 = U_n$ and $\mathcal{D}_2 = \{(1 \pm 2\varepsilon)/n\}$ (resp. $\{(1 \pm 4\varepsilon)/n\}$). Right: $\mathcal{D}_1 = U_2$ and $\mathcal{D}_2 = \{(1 \pm 2\varepsilon)/2\}$. The sequential tester stops as soon as the statistic enters the red region (for $H_1$) or blue region (for $H_2$) whereas the batch tester waits for the red and blue regions to cover the whole segment $[0, 1]$. The blue/red and black dashed lines represent respectively the stopping times of the sequential and batch algorithms. We note that, in both cases, the sequential tester stops long before the batch algorithm.

The proof is inspired by Karp and Kleinberg [2007] who proved lower bounds for testing whether the mean of a sequence of i.i.d. Bernoulli variables is smaller or larger than $1/2$. We construct well-chosen distributions $D_k$ (for $k$ integer) that are at distance $\varepsilon_k$ ($\varepsilon_k$ decreasing to 0) from uniform and then use properties of the Kullback-Leibler's divergence to show that no algorithm can distinguish between $D_k$ and uniform using fewer samples than in Eq. (2).

**Discussion on the setting and related work**  It is clearly impossible to test $\mathcal{D}_1 = \mathcal{D}_2$ versus $\mathcal{D}_1 \neq \mathcal{D}_2$ in finite time: this is why we introduce the slack parameter $\varepsilon$. Other authors like Daskalakis and Kawase [2017] make a different choice: they fix no $\varepsilon$, but only require that the test decides for $\mathcal{D}_1 \neq \mathcal{D}_2$ as soon as it can, and never stops with high probability when $\mathcal{D}_1 = \mathcal{D}_2$. We focus on the TV distance in testing closeness problems because it characterises the probability of error for the problem of distributions discrimination ; as noted by Daskalakis et al. [2018], using other distances such as KL and $\chi^2$ is in general impossible.

For an overview of testing discrete distributions we recommend the survey of Canonne [2020]. Testing closeness was solved by Chan et al. [2014], and a distribution dependent complexity was found by Diakonikolas and Kane [2016] and finally the high probability version by Diakonikolas et al. [2020]. Moreover, the problem of testing $\mathcal{D}_1 = \mathcal{D}_2$ vs $\mathcal{D}_1 \neq \mathcal{D}_2$ was solved by Daskalakis and Kawase [2017] for $n = 2$, however the constants are not optimal. They also propose algorithms for the general case using black-box reduction from non-sequential hypothesis testers. Sequential and adaptive procedures have also been explored in active hypothesis setting [Naghshvar and Javidi, 2013] and channels' discrimination [Hayashi, 2009]. Sequential strategies have been also considered for testing continuous distributions by Zhao et al. [2016] and Balsubramani and Ramdas [2015]. In the latter, the authors design sequential algorithms whose stopping time adapts to the unknown difficulty of the problem. The techniques used are time uniform concentration inequalities which are surveyed by Howard et al. [2020]. In contrast to the present work, however, they test on the *means* of the distributions.

## 2 Preliminaries

We follow mostly Daskalakis and Kawase [2017] in the notation. Given two distributions $\mathcal{D}_1$ and $\mathcal{D}_2$ on $\{1, \ldots, n\}$ we want to distinguish between two hypothesis $H_1 : \mathcal{D}_1 = \mathcal{D}_2$ and $H_2 : \mathrm{TV}(\mathcal{D}_1, \mathcal{D}_2) > \varepsilon$. We call a stopping rule a function $T : \bigcup_{k \in \mathbb{N}} [n]^k \times [n]^k \to \{0, 1, 2\}$ such that if $T(x, y) \neq 0$ then $T(xz, yt) = T(x, y)$ for all strings $x, y, z, t$ with $|x| = |y|$ and $|z| = |t|$.

$T(x, y) = 1$ (resp. $T(x, y) = 2$) means that the rule accepts $H_1$ (resp. $H_2$) after seeing the sequences $x$ and $y$ while $T(x, y) = 0$ means the rule doesn't make a choice and continue sampling. We define two different stopping times, the first $\tau_1(T, \mathcal{D}_1, \mathcal{D}_2) = \inf\{t, T(x_1 \cdots x_t, y_1 \ldots y_t) = 1\}$ and the second $\tau_2(T, \mathcal{D}_1, \mathcal{D}_2) = \inf\{t, T(x_1 \cdots x_t, y_1 \ldots y_t) = 2\}$ where $x_1, \ldots$ are i.i.d. samples from $\mathcal{D}_1$ and $y_1, \ldots$ samples from $\mathcal{D}_2$. We want to find stopping rules satisfying $\mathbb{P}\left(\tau_2(T, \mathcal{D}_1, \mathcal{D}_2) \leq \tau_1(T, \mathcal{D}_1, \mathcal{D}_2)\right) \leq \delta$ if $\mathcal{D}_1 = \mathcal{D}_2$ and $\mathbb{P}\left(\tau_1(T, \mathcal{D}_1, \mathcal{D}_2) \leq \tau_2(T, \mathcal{D}_1, \mathcal{D}_2)\right) \leq \delta$ whenever $\text{TV}(\mathcal{D}_1, \mathcal{D}_2) > \varepsilon$. We call such stopping rules $\delta$-correct. Our goal is to minimize the expected sample complexity $\mathbb{E}(\tau_1(T, \mathcal{D}_1, \mathcal{D}_2))$ in case of the input is from $\mathcal{D}_1, \mathcal{D}_2$ such that $\mathcal{D}_1 = \mathcal{D}_2$ and $\mathbb{E}(\tau_2(T, \mathcal{D}_1, \mathcal{D}_2))$ in case of the input is from $\mathcal{D}_1, \mathcal{D}_2$ such that $\text{TV}(\mathcal{D}_1, \mathcal{D}_2) > \varepsilon$.

# 3 Testing closeness for small $n$

In this section, we focus on small $n \geq 2$ and we consider two distributions $\mathcal{D}_1$ and $\mathcal{D}_2$ on $[n]$. We are testing two hypothesis $H_1$: $\mathcal{D}_1 = \mathcal{D}_2$ and $H_2$: $\text{TV}(\mathcal{D}_1, \mathcal{D}_2) > \varepsilon$. We are interested in precisely comparing the sample complexity of testing closeness in the sequential versus the batch setting. In order to find the optimal constant, we first need to obtain a sharp lower bound in the batch setting, which is done directly by using Stirling's approximation. We then turn to the sequential case.

## 3.1 Batch setting

In the batch setting, the number of steps $\tau$ is fixed before the test. The tester samples $A_1, \ldots, A_\tau \sim \mathcal{D}_1$ and $B_1, \ldots, B_\tau \sim \mathcal{D}_2$ then decides according to the comparison between the empirical TV distance $\text{TV}(\tilde{\mathcal{D}}_{1\tau}, \tilde{\mathcal{D}}_{2\tau})$ and $\varepsilon/2$ where $\tilde{\mathcal{D}}_{1\tau} = \left\{\left(\sum_{j=1}^{\tau} 1_{A_j=i}\right)/\tau\right\}_{i \in [n]}$ and $\tilde{\mathcal{D}}_{2\tau} = \left\{\left(\sum_{j=1}^{\tau} 1_{B_j=i}\right)/\tau\right\}_{i \in [n]}$ are the empirical distributions. If $\text{TV}(\tilde{\mathcal{D}}_{1\tau}, \tilde{\mathcal{D}}_{2\tau}) \leq \varepsilon/2$, she accepts $H_1$ and rejects it otherwise. In order to control the number of steps $\tau$ so that the error of this algorithm does not exceed $\delta$, McDiarmid's inequality (Habib et al. [2013]) writes for $\tau = \frac{4 \log(2^{n/2}/\delta)}{\varepsilon^2}$:

$$\mathbb{P}\left(\exists B \subset [n/2] : \left|\tilde{\mathcal{D}}_{1,\tau}(B) - \mathcal{D}_1(B) - \tilde{\mathcal{D}}_{2,\tau}(B) + \mathcal{D}_2(B)\right| > \frac{\varepsilon}{2}\right) \leq \sum_{B \subset [n/2]} e^{-\tau \varepsilon^2/4} \leq \delta . \quad \text{(M)}$$

Using the following property of TV distance: $\text{TV}(\mathcal{D}_1, \mathcal{D}_2) = \max_{B \subset [\lfloor n/2 \rfloor]} |\mathcal{D}_1(B) - \mathcal{D}_2(B)|$ along with the concentration inequality (M) for $\mathcal{D}_1 = \mathcal{D}_2$ (to control the type I error) and for $\mathcal{D}_1 \neq \mathcal{D}_2$ (to control the type II error) we prove that this test is $\delta$-correct. We show in the following proposition that this number of steps $\tau$ is necessary.

**Proposition 3.1.** *In the batch setting, the algorithm consisting of accepting $H_1$ when* $\text{TV}(\tilde{\mathcal{D}}_{1\tau}, \tilde{\mathcal{D}}_{2\tau}) \leq \varepsilon/2$ *and rejecting it otherwise is $\delta$-correct for $\tau = \frac{4 \log(2^{n/2}/\delta)}{\varepsilon^2}$.*

*Moreover, any $\delta$-correct algorithm testing closeness requires at least $\tau$ samples, where*

$$\tau \geq \min\left\{\frac{\log(1/2\delta)}{2 \text{KL}(1/2 - \varepsilon/4 \| 1/2 - \varepsilon/2)}, \frac{\log(1/2\delta)}{2 \text{KL}(1/2 + \varepsilon/4 \| 1/2)}\right\} - \mathcal{O}\left(\frac{\log \log(1/\delta)}{\varepsilon^2}\right) .$$

For this proof, we show that every $\delta$-correct tester can be transformed into a test which depends only on the numbers of $1's, 2's, \ldots, n's$ occurred on $\{A_1, \ldots, A_\tau\}$ and $\{B_1, \ldots, B_\tau\}$. We then consider the distributions $\mathcal{D}_{1,2} = \{1/2, 1/2, 0, \ldots, 0\}$ or $\mathcal{D}_{1,2} = \{1/2 \pm \varepsilon/2, 1/2 \mp \varepsilon/2, 0, \ldots, 0\}$ depending on the outcome of the algorithm when it sees two words of samples having respectively $\tau(1/2 - \varepsilon/4)$ and $\tau(1/2 + \varepsilon/4)$ ones (the rest of samples are equal to 2) and derive tight lower bounds on the probability mass function of the multinomial distribution (the full proof is deferred to App. A).

This simple analysis relies on well-known arguments for testing Bernoulli variables $\mathcal{D}_1 = \mathcal{B}(p)$ and $\mathcal{D}_2 = \mathcal{B}(q)$. For example, Anthony and Bartlett [2009] and Karp and Kleinberg [2007] test whether $q = 1/2 + \varepsilon$ or $q = 1/2 - \varepsilon$ with an error probability $\delta$. Anthony and Bartlett [2009] show that we need roughly $\log(1/\delta)\varepsilon^{-2}/4$ samples while Karp and Kleinberg [2007] prove that $2\log(1/\delta)\varepsilon^{-2}$ samples are sufficient. If $\varepsilon$ is not known to the tester, sequential algorithms prove to be essential. Indeed, Karp and Kleinberg [2007] manage to prove that $\Theta(\log\log(1/|q - 1/2|)|q - 1/2|^{-2})$ is necessary and sufficient to test $q > 1/2$ vs $q < 1/2$ with an error probability $1/3$. In what follows, we use sequential algorithms to expose the dependency on $\mathrm{TV}(\mathcal{D}_1, \mathcal{D}_2)$ for the testing closeness problem.

## 3.2   Sequential setting

If one wants to leverage the sequential setting to improve the optimal sample complexity of testing closeness, it is natural to first investigate how it can be improved by removing the batch assumption of the previous lower-bound in Proposition 3.1. We first state a new lower bound inspired from the work of Garivier and Kaufmann [2019].

**Proposition 3.2.** *Let $T$ be a stopping rule for testing closeness: $\mathcal{D}_1 = \mathcal{D}_2$ vs $\mathrm{TV}(\mathcal{D}_1, \mathcal{D}_2) > \varepsilon$ with an error probability $\delta$. Let $\tau_1$ and $\tau_2$ the associated stopping times. We have*

$$\sup_{\mathcal{D}_1 = \mathcal{D}_2} \mathbb{E}(\tau_1(\mathcal{D}_1, \mathcal{D}_2)) \geq \frac{\log(1/3\delta)}{\mathrm{KL}(1/2\|1/2 + \varepsilon/2) + \mathrm{KL}(1/2\|1/2 - \varepsilon/2)} \underset{\varepsilon \to 0}{\sim} \frac{\log(1/3\delta)}{\varepsilon^2} \ and$$

$$\sup_{\mathrm{TV}(\mathcal{D}_1, \mathcal{D}_2) = d} \mathbb{E}(\tau_2(\mathcal{D}_1, \mathcal{D}_2)) \geq \frac{\log(1/3\delta)}{\mathrm{KL}(1/2 + d/2\|1/2) + \mathrm{KL}(1/2 - d/2\|1/2)} \underset{d \to 0}{\sim} \frac{\log(1/3\delta)}{d^2} \ if \ d > \varepsilon \ .$$

An average number of samples equivalent to $\log(1/3\delta)(\varepsilon \vee \mathrm{TV}(\mathcal{D}_1, \mathcal{D}_2))^{-2}$ is thus necessary when the tester can access sequentially to the samples, which is roughly 4 times less than the complexity obtained in the batch setting. The proof, with a strong information-theoretic flavor, compares two situations: when the samples are from equal distributions and when they are from $\varepsilon$-far distributions. Those samples cannot be distinguished until their size is large enough, as can be proved by combining properties of Kullback-Leibler's divergence and Wald's lemma (see the detailed proof in App. A).

In the sequential testing, the tester chooses when to stop according to the previous observations $((A_1, B_1), \ldots, (A_t, B_t))$, making comparisons at each step $t$. The key explanation of the sequential speedup is that the tester can stop as soon as she is sure that she can accept one of the hypothesis $H_1$ or $H_2$. On the contrary, in the batch setting she had to sample enough observation to be simultaneously sure that either $H_1$ or $H_2$ hold. In this aim, at each time step, after sampling a new observation $(A_t, B_t)$, she compares the updated empirical TV distance $S_t = \mathrm{TV}(\tilde{\mathcal{D}}_{1t}, \tilde{\mathcal{D}}_{2t})$ to specific thresholds and sees if (a) $S_t$ is sufficiently far from 0 to surely accept $H_2$, (b) $S_t$ is sufficiently close to $\varepsilon$ to surely accept $H_1$, (c) she is unsure and needs further samples to take a sound decision. This test is formally described in Alg. 1 and its execution is illustrated in Figure 1 for $n = 2$.

To show the correctness of such sequential algorithms, we need here no more than McDiarmid's inequality (M) and the union bound:

$$\mathbb{P}\left(\exists t \geq 1, \exists B \subset [n/2] : \left|\tilde{\mathcal{D}}_{1,t}(B) - \mathcal{D}_1(B) - \tilde{\mathcal{D}}_{2,t}(B) + \mathcal{D}_2(B)\right| > \Phi_t\right) \leq \delta,$$

where $\Phi_t$ denote the constant $\Phi_t = \sqrt{\log\left(\frac{2^{n-1}t(t+1)}{\delta}\right)/t}$. On the other hand, to control the sample complexity, we prove upper bounds on the expected stopping times:

$$\tau_1 = \inf\left\{t \geq 1 : \mathrm{TV}\left(\tilde{\mathcal{D}}_{1,t}, \tilde{\mathcal{D}}_{2,t}\right) \leq \varepsilon - \Phi_t\right\}, \ \text{and} \ \tau_2 = \inf\left\{t \geq 1 : \mathrm{TV}\left(\tilde{\mathcal{D}}_{1,t}, \tilde{\mathcal{D}}_{2,t}\right) > \Phi_t\right\}.$$

---

**Algorithm 1** Distinguish between $\mathcal{D}_1 = \mathcal{D}_2$ and $\mathrm{TV}(\mathcal{D}_1, \mathcal{D}_2) > \varepsilon$ with high probability

---

**Require:** $A_1, \ldots$ samples from $\mathcal{D}_1$ and $B_1, \ldots$ samples from $\mathcal{D}_2$
**Ensure:** Accept if $\mathcal{D}_1 = \mathcal{D}_2$ and Reject if $\mathrm{TV}(\mathcal{D}_1, \mathcal{D}_2) > \varepsilon$ with probability of error less than $\delta$

    $t = 1, W = 1$
    **while** $W = 1$ **do**
       $\tilde{\mathcal{D}}_{1,t} = \left\{ \left( \sum_{j=1}^{t} \mathbb{1}_{A_j=i} \right) / t \right\}_{i \in [n]}, \tilde{\mathcal{D}}_{2,t} = \left\{ \left( \sum_{j=1}^{t} \mathbb{1}_{B_j=i} \right) / t \right\}_{i \in [n]}$

       **if** $\mathrm{TV}\left( \tilde{\mathcal{D}}_{1,t}, \tilde{\mathcal{D}}_{2,t} \right) > \sqrt{\dfrac{\log\left( \frac{2^{n-1} t(t+1)}{\delta} \right)}{t}}$ **then**
          $W = 0$
          **return** Reject

       **else if** $\mathrm{TV}\left( \tilde{\mathcal{D}}_{1,t}, \tilde{\mathcal{D}}_{2,t} \right) \le \varepsilon - \sqrt{\dfrac{\log\left( \frac{2^{n-1} t(t+1)}{\delta} \right)}{t}}$ **then**
          $W = 0$
          **return** Accept
       **else**
          $t = t + 1$
       **end if**
    **end while**

---

It is clear that the stopping time of the algorithm is random. Yet, we can show that this algorithm stops before the non sequential one and give an upper bound on the expected stopping time $\tau$ (or expected sample complexity) using the inequality $\mathbb{E}(\tau) \le N + \sum_{t \ge N} \mathbb{P}(\tau \ge t)$, where $N$ is chosen so that $\mathbb{P}(\tau \ge t)$ is (exponentially) small for $t \ge N$ (see Lemma 4.3). In the following theorem, we state an upper bound on the estimated sample complexity of this algorithm. The full analysis of this algorithm is detailed in App. B.

**Theorem 3.3.** *The Alg. 1 is $\delta$-correct and its stopping times verify for $n \le \mathcal{O}(\log(1/\delta)^{1/3})$:*

$$\mathbb{E}(\tau_1(\mathcal{D}, \mathcal{D})) \le \frac{\log(2^{n+1}/\delta)}{\varepsilon^2} + \mathcal{O}\left( \frac{\log(2^{n+1}/\delta)^{2/3}}{\varepsilon^2} \right) \text{ if } \mathcal{D}_1 = \mathcal{D}_2 = \mathcal{D} \text{ and}$$

$$\mathbb{E}(\tau_2(\mathcal{D}_1, \mathcal{D}_2)) \le \frac{\log(2^{n+1}/\delta)}{\mathrm{TV}(\mathcal{D}_1, \mathcal{D}_2)^2} + \mathcal{O}\left( \frac{\log(2^{n+1}/\delta)^{2/3}}{\mathrm{TV}(\mathcal{D}_1, \mathcal{D}_2)^2} \right) \text{ if } \mathrm{TV}(\mathcal{D}_1, \mathcal{D}_2) > \varepsilon .$$

These upper bounds are tight on the sense that they match the asymptotic lower bounds of Lemma 3.2 if $n \ll \log(1/\delta)$. We see here the many advantages of the sequential setting: (a) the sequential algorithm stops always before the non-sequential algorithm since after the batch complexity the decision regions of the sequential algorithm intersect, (b) the estimated sample complexity is $4$ times less than the optimal complexity in the non sequential setting, (c) the sample complexity in the sequential setting depends on the unknown distributions $\mathcal{D}_1$ and $\mathcal{D}_2$ through the distance $\mathrm{TV}(\mathcal{D}_1, \mathcal{D}_2)$. Note that this cannot be the case in the batch setting as the number of sample should be fixed beforehand. This attribute makes a considerable difference when $\mathcal{D}_1$ is very different from $\mathcal{D}_2$. Nevertheless, the above lower bounds and upper bounds do not match exactly, the dependence on $n$ cannot be avoided if $n$ is of the order (or larger) of $\log(1/\delta)$. Finally, we can always truncate a sequential algorithm with the batch one in a way to get the best sample complexity in every regime.

## 4 Testing closeness-the general case

We consider here the general case $n \ge 3$. We recall that we have two unknown distributions $\mathcal{D}_1$ and $\mathcal{D}_2$ and want to distinguish between $\mathcal{D}_1 = \mathcal{D}_2$ and $\mathrm{TV}(\mathcal{D}_1, \mathcal{D}_2) > \varepsilon$ with high probability $1 - \delta$. Inspired by the small-$n$ case, we describe how to transform a batch to a sequential algorithm with better expected sample complexity. In that case, exactly identifying the sample complexity remains out of reach: the dependency on $\varepsilon$, $\delta$ and $n$ can be computed only up to a multiplicative constant.

## 4.1 Batch setting

Recently, Diakonikolas et al. [2020] have shown that the dependence on the error probability in the sample complexity of the closeness problem could be better than the $\log 1/\delta$ found by repeating $\log 1/\delta$ times the classical algorithm of Chan et al. [2014] and accepting or rejecting depending on the majority test. More precisely:

**Theorem 4.1** (Diakonikolas et al. [2020]). $\Theta\left(\max\left(\frac{n^{2/3}\log^{1/3}(1/\delta)}{\varepsilon^{4/3}}, \frac{n^{1/2}\log^{1/2}(1/\delta)}{\varepsilon^2}, \frac{\log(1/\delta)}{\varepsilon^2}\right)\right)$ *samples are necessary and sufficient to test whether* $\mathcal{D}_1 = \mathcal{D}_2$ *or* $\mathrm{TV}(\mathcal{D}_1, \mathcal{D}_2) > \varepsilon$ *with an error* $\delta > 0$.

The main ingredient of a closeness tester is an efficient test statistic which can distinguish between the two hypothesis. Let us define by $X_i$ (resp. $Y_i$) the number of samples from $\mathcal{D}_1$ (resp. $\mathcal{D}_2$) whose values are equal to $i \in [n]$. Thinking to the TV distance we use, we could be tempted to take a decision based on the statistic $\sum_{i=1}^n |X_i - Y_i|$. However this simple statistic suffers from a principal caveat: its expected value is neither zero neither easily lower bounded when $\mathcal{D}_1 = \mathcal{D}_2$. As a remedy, Diakonikolas et al. [2020] propose to use the following statistic: $Z = \sum_{i=1}^n |X_i - Y_i| + |X_i' - Y_i'| - |X_i - X_i'| - |Y_i - Y_i'|$, where $X_i'$ and $Y_i'$ correspond to a second set of independent samples. The expected value of the estimator $Z$ is obviously 0 when the distributions $\mathcal{D}_1$ and $\mathcal{D}_2$ are equal. On the other hand when $\mathrm{TV}(\mathcal{D}_1, \mathcal{D}_2) > \varepsilon$, they provide a lower bound on the expected value of the estimator $Z$ which enable to test closeness between $\mathcal{D}_1$ and $\mathcal{D}_2$. Since these results turn out to be similarly useful in our subsequent analysis, we summarized them in the following lemma.

**Lemma 4.2** (Diakonikolas et al. [2020]). *Let* $d = \mathrm{TV}(\mathcal{D}_1, \mathcal{D}_2)$. *Let* $k \geq 1$ *and* $(k_1, k_2, k_1', k_2') \sim Multinom(4k, (1/4, 1/4, 1/4, 1/4))$. *Let* $(X_i)_{i=1}^{k_1}$ *and* $(X_i')_{i=1}^{k_1'}$ *two sets of i.i.d. samples from* $\mathcal{D}_1$ *and* $(Y_i)_{i=1}^{k_2}$ *and* $(Y_i')_{i=1}^{k_2'}$ *two sets of independent samples from* $\mathcal{D}_2$. *Then there are universal constants* $c$ *and* $C$ *such that*

- *If* $\mathcal{D}_1 = \mathcal{D}_2$, $\mathbb{E}[Z] = 0$.
- *If* $\mathrm{TV}(\mathcal{D}_1, \mathcal{D}_2) > \varepsilon$, $\mathbb{E}[Z] \geq C \min\left\{kd, \frac{k^2 d^2}{n}, \frac{k^{3/2} d^2}{\sqrt{n}}\right\} - c\sqrt{k}$.

The lower bound on the expectation of $Z$ is obtained by Poissonization. This bound is stronger than the one obtained for the *chi*-square estimator; $\sum_{i=1}^n \frac{(X_i - Y_i)^2 - X_i - Y_i}{X_i + Y_i}$, used by Chan et al. [2014]. Indeed for far distributions the lower bound on the expected value of the *chi*-square estimator does not allow the best dependency on $\varepsilon$ and $\delta$. This lemma is the key ingredient behind the batch algorithm. Indeed, for sufficiently large $k = \Omega\left(\max\left(\frac{n^{2/3}\log^{1/3}(1/\delta)}{\varepsilon^{4/3}}, \frac{n^{1/2}\log^{1/2}(1/\delta)}{\varepsilon^2}, \frac{\log(1/\delta)}{\varepsilon^2}\right)\right)$, Diakonikolas et al. [2020] show that $\mathbb{E}[Z] \geq C''\sqrt{k\log 1/\delta}$ for a universal constant $C''$ if $\mathrm{TV}(\mathcal{D}_1, \mathcal{D}_2) > \varepsilon$, then by applying McDiarmid's inequality they prove that the algorithm consisting of returning $H_2$ if $Z \geq C''\sqrt{k\log 1/\delta}/2$ and returning $H_1$ otherwise is $\delta$-correct. In the following we draw our inspiration from their work to design a sequential algorithm for testing closeness.

## 4.2 Sequential setting

We present here how the sequential setting can improve the sample complexity found in the batch setting. Our sequential tester is based on the test statistic $Z$ of Diakonikolas et al. [2020], but we allow the stopping rules of this new algorithm to be time-dependent. When the distributions to be tested $\mathcal{D}_1$ and $\mathcal{D}_2$ are equal, the estimator $Z_t$ cannot be very large, and if they are $\varepsilon$-far, the estimator cannot be very small: at each step, the tester compares $Z_t$ to some well chosen thresholds. If she cannot decide with sufficient confidence, she asks for more samples. This can possibly last until the two decision regions meet. This time has the order of the complexity of the batch algorithm. This tester is formally

---

**Algorithm 2** Distinguish between $\mathcal{D}_1 = \mathcal{D}_2$ and $\mathrm{TV}(\mathcal{D}_1, \mathcal{D}_2) > \varepsilon$ with high probability

---

**Require:** $A_1, \ldots$ samples from $\mathcal{D}_1$ and $B_1, \ldots$ samples from $\mathcal{D}_2$
**Ensure:** Accept if $\mathcal{D}_1 = \mathcal{D}_2$ and Reject if $\mathrm{TV}(\mathcal{D}_1, \mathcal{D}_2) > \varepsilon$ with probability of error less than $\delta$
  $t = 1, W = 1$
  **while** $W = 1$ **do**
    $(m_{1,t}, m'_{1,t}, m_{2,t}, m'_{2,t}) \sim Multinom(4t, (1/4, 1/4, 1/4, 1/4))$

$$Z_t = \sum_{i=1}^{n} |X_i - Y_i| + |X'_i - Y'_i| - |X_i - X'_i| - |Y_i - Y'_i| \,,$$

    where $X_i$(resp. $X'_i, Y_i, Y'_i$) are the numbers of $i$'s in the word formed with $m_{1,t}$ (resp. $m'_{1,t}, m_{2,t}, m'_{2,t}$) samples from $\mathcal{D}_1$ (resp. $\mathcal{D}_1, \mathcal{D}_2, \mathcal{D}_2$) . We need only to sample the difference of $(m_{1,t} - m_{1,t-1})^+ + (m'_{1,t} - m'_{1,t-1})^+$ from $\mathcal{D}_1$ and $(m_{2,t} - m_{2,t-1})^+ + (m'_{2,t} - m'_{2,t-1})^+$ from $\mathcal{D}_2$.
    **if** $|Z_t| > 2\sqrt{2t \log\left(\frac{\pi^2}{3\delta}\right) + 4et \log(\log(t) + 1)}$ **then**
      $W = 0$
      **return** Reject
    **else if** $|Z_t| \leq C \min\left\{t\varepsilon, \frac{t^2\varepsilon^2}{n}, \frac{t^{3/2}\varepsilon^2}{\sqrt{n}}\right\} - c\sqrt{t} - 2\sqrt{2t \log\left(\frac{\pi^2}{3\delta}\right) + 4et \log(\log(t) + 1)}$ **then**
      $W = 0$
      **return** Accept
    **else**
      $t = t + 1$
    **end if**
  **end while**

---

defined in Alg. 2. For sake of simplicity, let us denote by $\Delta_t = C \min\left\{t\varepsilon, \frac{t^2\varepsilon^2}{n}, \frac{t^{3/2}\varepsilon^2}{\sqrt{n}}\right\} - c\sqrt{t}$ and by $\Psi_t = 2\sqrt{2t \log\left(\frac{\pi^2}{3\delta}\right) + 4et \log(\log(t) + 1)}$. The stopping times $\tau_1$ and $\tau_2$ of Alg. 2 are then defined by

$$\tau_1 = \inf\{t \geq 1 : |Z_t| \leq \Delta_t - \Psi_t\}, \text{ and } \tau_2 = \inf\{t \geq 1 : |Z_t| > \Psi_t\} \,.$$

We prove now that Alg. 2 is $\delta$-correct and then study its sample complexity.

### 4.2.1 Correctness

We prove here that Alg. 2 has an error probability less than $\delta$. The proof relies on the following uniform concentration lemma for $Z_t$:

**Lemma 4.3.** *For $\eta, s > 1$, let $J(\eta, s, t) = \sqrt{2\eta ts \log\left(\frac{\log(t)}{\log(\eta)} + 1\right) - 2t \log(\zeta(s)^{-1}\delta/2)}$ , where $\zeta(s) = \sum_{n \geq 1} \frac{1}{n^s}$. Then*

$$\mathbb{P}\left(\exists t \geq 1 : |Z_t - \mathbb{E}[Z_t]| > J(\eta, s, 4t)\right) \leq \delta \,.$$

The proof of this lemma is inspired from Howard et al. [2018] and relies on dividing the set of integers into some well chosen subsets, applying union bound and finally invoking McDiarmid's inequality with specific arguments for each interval. It is deferred to Appendix F.4. Note that Lemma 4.3 yields the best second order term in the complexity, in contrast to a simple union bound on McDiarmid's inequality. This feature is not essential for the study of the testing closeness problem as we are interested here in leading terms only (see Theorem 4.4). However, the $\log - \log$ dependency proves useful when showing that Alg. 2 used with $\varepsilon = 0$ obtains the optimal sample complexity for

testing $\mathcal{D}_1 = \mathcal{D}_2$ vs $\mathcal{D}_1 \neq \mathcal{D}_2$ (see Theorem 4.6). For $\eta = e$ and $s = 2$, the function $J$ becomes $J(e, 2, 4t) = \Psi_t$ and Lemma 4.3 proves the correctness of Alg. 2 as sketched below:

- If $\mathcal{D}_1 = \mathcal{D}_2$, the probability of error is bounded as $\mathbb{P}\left(\tau_2 \leq \tau_1\right) \leq \mathbb{P}\left(\exists t \geq 1 : |Z_t| > \Psi_t\right) \leq \delta$.
- If $\mathrm{TV}(\mathcal{D}_1, \mathcal{D}_2) > \varepsilon$, the probability of error can be bounded as:

$$\mathbb{P}\left(\tau_1 \leq \tau_2\right) = \mathbb{P}\left(\exists t \geq 1 : |Z_t| \leq \Delta_t - \Psi_t\right) \overset{(i)}{\leq} \mathbb{P}\left(\exists t \geq 1 : |Z_t - \mathbb{E}(Z_t)| \geq \mathbb{E}(Z_t) - \Delta_t + \Psi_t\right)$$

$$\overset{(ii)}{\leq} \mathbb{P}\left(\exists t \geq 1 : |Z_t - \mathbb{E}(Z_t)| \geq \Psi_t\right) \overset{(iii)}{\leq} \delta.$$

where $(i)$ follows from the triangular inequality $|Z_t - \mathbb{E}(Z_t)| \geq \mathbb{E}(Z_t) - Z_t$, $(ii)$ follows by the fact that $\mathbb{E}(Z_t) \geq \Delta_t$ from Lemma 4.2 and $(iii)$ follows from Lemma 4.3.

### 4.2.2 Complexity

The following theorem shows the preeminence of our sequential algorithm, by bounding the expectations of the stopping times $\tau_1$ and $\tau_2$.

**Theorem 4.4.** *Let $d = \mathrm{TV}(\mathcal{D}_1, \mathcal{D}_2)$. The sample complexity of Alg. 2 satisfies*

- *If $\mathcal{D}_1 = \mathcal{D}_2$, $\mathbb{E}(\tau_1(T, \mathcal{D}_1, \mathcal{D}_2)) \leq 2N_\varepsilon$.*
- *If $\mathrm{TV}(\mathcal{D}_1, \mathcal{D}_2) > \varepsilon$, $\mathbb{E}(\tau_2(T, \mathcal{D}_1, \mathcal{D}_2)) \leq 2N_d$.*

*where for all $\eta > 0$, $N_\eta$ is defined by*

$$N_\eta = \max\left\{ \frac{128}{C^2} \frac{\log(\frac{\pi^2}{3\delta})}{\eta^2} + \frac{512e}{C^2\eta^2} \log\left(\log\left(\frac{128\log(\frac{\pi^2}{3\delta})}{\eta^2 C^2}\right) + 1\right) + \frac{16c^2}{C^2\eta^2}, \right.$$

$$\left(\frac{128}{C^2} \frac{n^2\log(\frac{\pi^2}{3\delta})}{\eta^4} + \frac{512en^2}{C^2\eta^4} \log\left(\log\left(\frac{128}{C^2} \frac{n^2\log(\frac{\pi^2}{3\delta})}{\eta^4}\right) + 1\right) + \frac{16c^2n^2}{\eta^4 C^2}\right)^{1/3},$$

$$\left.\left(\frac{128}{C^2} \frac{n\log(\frac{\pi^2}{3\delta})}{\eta^4} + \frac{512en}{C^2\eta^4} \log\left(\log\left(\frac{128}{C^2} \frac{n\log(\frac{\pi^2}{3\delta})}{\eta^4}\right) + 1\right) + \frac{16c^2n}{\eta^4 C^2}\right)^{1/2}\right\},$$

*and the constants $c$ and $C$ come from Lemma 4.2.*

This theorem states that $\mathcal{O}\left(\max\left(\frac{n^{2/3}\log^{1/3}(1/\delta)}{(\varepsilon \vee \mathrm{TV}(\mathcal{D}_1, \mathcal{D}_2))^{4/3}}, \frac{n^{1/2}\log^{1/2}(1/\delta)}{(\varepsilon \vee \mathrm{TV}(\mathcal{D}_1, \mathcal{D}_2))^2}, \frac{\log(1/\delta)}{(\varepsilon \vee \mathrm{TV}(\mathcal{D}_1, \mathcal{D}_2))^2}\right)\right)$ samples are sufficient to distinguish between $\mathcal{D}_1 = \mathcal{D}_2$ and $\mathrm{TV}(\mathcal{D}_1, \mathcal{D}_2) > \varepsilon$ with high probability and its proof can be found in App. C. We remark that after $N_\varepsilon$ steps, the two stopping conditions of Alg. 2 cannot be both unsatisfied. Therefore, the Alg. 2 stops surely before $N_\varepsilon$ hence it has at least a comparable complexity, in the leading terms, of the batch algorithm of Diakonikolas et al. [2020] when $\mathcal{D}_1 = \mathcal{D}_2$. Moreover, Alg. 2 has the advantage of stopping rapidly when $\mathcal{D}_1$ and $\mathcal{D}_2$ are far away. It turns out that Alg. 2 is tight, the following lower bounds show that its complexity is optimal up to multiplicative constant.

**Theorem 4.5.** *There is no stopping rule $T$ for the problem of testing $\mathcal{D}_1 = \mathcal{D}_2$ vs $\mathrm{TV}(\mathcal{D}_1, \mathcal{D}_2) > \varepsilon$ with an error probability $\delta$ such that*

$$\mathbb{P}\left(\tau_2(T, \mathcal{D}_1, \mathcal{D}_2) \leq c\frac{n^{1/2}\log(1/3\delta)^{1/2}}{\mathrm{TV}(\mathcal{D}_1, \mathcal{D}_2)^2}\right) \geq 1 - \delta \ \text{ if } \mathrm{TV}(\mathcal{D}_1, \mathcal{D}_2) > \varepsilon \text{ and}$$

$$\mathbb{P}\left(\tau_1(T, \mathcal{D}_1, \mathcal{D}_2) \leq c\frac{n^{1/2}\log(1/3\delta)^{1/2}}{\varepsilon^2}\right) \geq 1 - \delta \ \text{ if } \mathcal{D}_1 = \mathcal{D}_2,$$

*where $c$ a universal constant. We have similar statement if we replace $\frac{n^{1/2}\log(1/3\delta)^{1/2}}{(\varepsilon \vee \mathrm{TV}(\mathcal{D}_1, \mathcal{D}_2))^2}$ by $\frac{\log(1/3\delta)}{(\varepsilon \vee \mathrm{TV}(\mathcal{D}_1, \mathcal{D}_2))^2}$ or $\frac{n^{2/3}\log(1/3\delta)^{1/3}}{(\varepsilon \vee \mathrm{TV}(\mathcal{D}_1, \mathcal{D}_2))^{4/3}}$.*

These results imply that if we are looking at bounds that depend only on $n, \epsilon$ and $\delta$, then sequential algorithms can at most gain a constant factor in terms of complexity. But it is possible that sequential algorithms stop much faster for some classes of distributions, such as if $\mathrm{TV}(\mathcal{D}_1, \mathcal{D}_2)$ is large. Investigating other classes of distributions for which we get an advantage is an interesting future direction. This theorem is proven using almost the same construction of distributions as for the batch lower bounds. We point out that the constructed distributions have a TV distance equal exactly to $d > \varepsilon$ instead of $\varepsilon$ and we use Wald's lemma along with the tensorization property of KL to deduce lower bounds on a random stopping time. The complete proof is deferred to App. D.

### 4.3 Special case, taking $\varepsilon \to 0$

If we take the precision $\varepsilon = 0$, Alg. 2 provides stopping rules for which the algorithm does not stop if $\mathcal{D}_1 = \mathcal{D}_2$ and rejects if $\mathcal{D}_1 \neq \mathcal{D}_2$ with probability at least $1 - \delta$. Hence, the upper bound on the stopping time $\tau_2$ can be translated to an upper bound on the stopping time of testing $\mathcal{D}_1 = \mathcal{D}_2$ vs $\mathcal{D}_1 \neq \mathcal{D}_2$. Theorem 4.4 and its proof in App. C show that, with high probability, we have $\tau_2 \leq N_{\mathrm{TV}(\mathcal{D}_1, \mathcal{D}_2)}$ and this upper bound is $\mathcal{O}\left(\frac{\log\log(1/d)}{d^2} \vee \frac{n^{2/3} \log\log(1/d)^{1/3}}{d^{4/3}} \vee \frac{n^{1/2} \log\log(1/d)^{1/2}}{d^2}\right)$ when $d = \mathrm{TV}(\mathcal{D}_1, \mathcal{D}_2) \to 0$. This is the object of the following theorem.

**Theorem 4.6.** *There is a stopping rule that can decide $\mathcal{D}_1 \neq \mathcal{D}_2$ with probability at least $9/10$ using at most $\mathcal{O}\left(\frac{\log\log(1/d)}{d^2} \vee \frac{n^{2/3} \log\log(1/d)^{1/3}}{d^{4/3}} \vee \frac{n^{1/2} \log\log(1/d)^{1/2}}{d^2}\right)$ samples where $d = \mathrm{TV}(\mathcal{D}_1, \mathcal{D}_2)$.*

This result improves upon the sample complexity of Daskalakis and Kawase [2017] where the dependency in $n$ is $n/\log n$. Furthermore, it is optimal. Indeed we cannot find stopping rules whose sample complexity is tighter than this upper bound as stated in the following theorem (see proof in App. E).

**Theorem 4.7.** *There is no stopping rule $T$ for the problem of testing $\mathcal{D}_1 = \mathcal{D}_2$ vs $\mathcal{D}_1 \neq \mathcal{D}_2$ with an error probability $1/10$ such that*

$$\mathbb{P}\left(\tau_2(T, \mathcal{D}_1, \mathcal{D}_2) \leq C \frac{n^{1/2} \log\log(1/d)^{1/2}}{d^2}\right) \geq \frac{15}{16},$$

*where $d = \mathrm{TV}(\mathcal{D}_1, \mathcal{D}_2)$ and $C$ a universal constant. We have similar statements if we replace $\frac{n^{1/2} \log\log(1/d)^{1/2}}{d^2}$ by $\frac{\log\log(1/d)}{d}$ or $\frac{n^{2/3} \log\log(1/d)^{1/3}}{d^{4/3}}$.*

To sum up, a number $\Theta\left(\frac{\log\log(1/d)}{d^2} \vee \frac{n^{2/3} \log\log(1/d)^{1/3}}{d^{4/3}} \vee \frac{n^{1/2} \log\log(1/d)^{1/2}}{d^2}\right)$ of samples is necessary and sufficient to decide whether $\mathcal{D}_1 = \mathcal{D}_2$ or $\mathcal{D}_1 \neq \mathcal{D}_2$ with probability $9/10$.

## 5 Conclusion

We have provided a tight analysis of the complexity of testing closeness for small $n$, where the importance of sequential procedures is clearly exhibited. We would like to emphasize that similar arguments permit to obtain sequential procedures for testing identity problem with a similar factor $4$ improvement.

For the general case, we proposed a tight algorithm for testing closeness where the complexity can depend on the actual TV distance between the two distributions. Our techniques can also be transferred to the testing identity problem based on the batch algorithm derived by Diakonikolas et al. [2017]. We note that for some specific families of distributions the improvement can be much more than the general one. This is the case of distributions concentrated in small sets which can be tested rapidly by sequential strategies and a direction worth pursuing.

