# References

Constantinos Daskalakis and Yasushi Kawase. Optimal stopping rules for sequential hypothesis testing. In *25th Annual European Symposium on Algorithms (ESA)*, 2017.

Ilias Diakonikolas, Themis Gouleakis, Daniel M Kane, John Peebles, and Eric Price. Optimal testing of discrete distributions with high probability. *arXiv preprint arXiv:2009.06540*, 2020.

Steven R Howard, Aaditya Ramdas, Jon McAuliffe, and Jasjeet Sekhon. Uniform, nonparametric, non-asymptotic confidence sequences. *arXiv preprint arXiv:1810.08240*, 2018.

Steven R Howard, Aaditya Ramdas, Jon McAuliffe, Jasjeet Sekhon, et al. Time-uniform chernoff bounds via nonnegative supermartingales. *Probability Surveys*, 17:257–317, 2020.

Richard M Karp and Robert Kleinberg. Noisy binary search and its applications. In *Proceedings of the eighteenth annual ACM-SIAM symposium on Discrete algorithms*, pages 881–890, 2007.

Constantinos Daskalakis, Gautam Kamath, and John Wright. Which distribution distances are sublinearly testable? In *Proceedings of the Twenty-Ninth Annual ACM-SIAM Symposium on Discrete Algorithms*, pages 2747–2764. SIAM, Philadelphia, PA, 2018. doi: 10.1137/1.9781611975031.175. URL https://doi.org/10.1137/1.9781611975031.175.

Clément L Canonne. A survey on distribution testing: Your data is big. but is it blue? *Theory of Computing*, pages 1–100, 2020.

Siu-On Chan, Ilias Diakonikolas, Paul Valiant, and Gregory Valiant. Optimal algorithms for testing closeness of discrete distributions. In *Proceedings of the twenty-fifth annual ACM-SIAM symposium on Discrete algorithms*, pages 1193–1203. SIAM, 2014.

Ilias Diakonikolas and Daniel M Kane. A new approach for testing properties of discrete distributions. In *2016 IEEE 57th Annual Symposium on Foundations of Computer Science (FOCS)*, pages 685–694. IEEE, 2016.

Mohammad Naghshvar and Tara Javidi. Sequentiality and adaptivity gains in active hypothesis testing. *IEEE Journal of Selected Topics in Signal Processing*, 7(5):768–782, 2013.

Masahito Hayashi. Discrimination of two channels by adaptive methods and its application to quantum system. *IEEE transactions on information theory*, 55(8):3807–3820, 2009.

Shengjia Zhao, Enze Zhou, Ashish Sabharwal, and Stefano Ermon. Adaptive concentration inequalities for sequential decision problems. *Advances in Neural Information Processing Systems*, 29: 1343–1351, 2016.

Akshay Balsubramani and Aaditya Ramdas. Sequential nonparametric testing with the law of the iterated logarithm. *arXiv preprint arXiv:1506.03486*, 2015.

Michel Habib, Colin McDiarmid, Jorge Ramirez-Alfonsin, and Bruce Reed. *Probabilistic methods for algorithmic discrete mathematics*, volume 16. Springer Science & Business Media, 2013.

Martin Anthony and Peter L Bartlett. *Neural network learning: Theoretical foundations*. cambridge university press, 2009.

Aurélien Garivier and Emilie Kaufmann. Non-asymptotic sequential tests for overlapping hypotheses and application to near optimal arm identification in bandit models. *arXiv preprint arXiv:1905.03495*, 2019.

Ilias Diakonikolas, Themis Gouleakis, John Peebles, and Eric Price. Optimal identity testing with high probability. *arXiv preprint arXiv:1708.02728*, 2017.

C Leubner. Generalised stirling approximations to n! *European Journal of Physics*, 6(4):299, 1985.

Abraham Wald. On cumulative sums of random variables. *The Annals of Mathematical Statistics*, 15 (3):283–296, 1944.


# A   General lower bounds and their proofs

In this section we present lower bounds for testing closeness in the general case of $n \geq 2$ and provide the proofs of the lower bounds presented in the paper.

## A.1   Proof of Theorem 3.1

We consider distributions supported only on $\{1, 2\}$, this is possible since we want that our algorithm would work for all distributions. We consider such a $\delta$-correct test $A : \{1, 2\}^\tau \times \{1, 2\}^\tau \to \{0, 1\}$, it sees two words consisting of $\tau$ samples either from equal distributions or $\varepsilon$-far ones and returns 0 if it thinks they are equal and 1 otherwise. We construct another test $B : \{1, 2\}^\tau \times \{1, 2\}^\tau \to \{0, 1\}$ by the expression

$$B(x, y) = 1_{\sum_{\sigma, \rho \in \mathcal{S}_\tau} A(\sigma(x), \rho(y)) \geq (\tau!)^2/2} \ ,$$

$B$ can be proven to be $2\delta$-correct and have the property of invariance under the action of the symmetric group. This leads to an algorithm $C : \{0, \dots, \tau\}^2 \to \{0, 1\}$ which is $2\delta$ correct and satisfies

$$C(i, j) = B(x_i, y_j) \ ,$$

where $x_k = 1 \dots 1 2 \dots 2$ with $k$ ones. We consider $i = [\tau(1/2 - \varepsilon/4)]$ and $j = [\tau(1/2 + \varepsilon/4)]$. We denote by $N_i(x)$ the number of $i$ in a word $x$ of length $\tau$ for $i = 1, 2$.

- If $C(i, j) = 0$, let $x$ (resp. $y$) a word of length $\tau$ constituted of i.i.d samples from $\{1/2 - \varepsilon/2, 1/2 + \varepsilon/2, 0, \dots, 0\}$ (resp. $\{1/2 + \varepsilon/2, 1/2 - \varepsilon/2, 0, \dots, 0\}$), then $\mathbb{P}_{1/2-\varepsilon/2, 1/2+\varepsilon/2}(N_1(x) = i, N_1(y) = j) \leq 2\delta$ hence with Stirling's approximation (Leubner [1985] )

$$\frac{e^{-2}}{2\pi\tau} e^{-\tau \operatorname{KL}(i/\tau \| 1/2 - \varepsilon/2)} e^{-\tau \operatorname{KL}(1-j/\tau \| 1/2 - \varepsilon/2)} \leq 2\delta.$$

Thus

$$2\tau \operatorname{KL}(1/2 + \varepsilon/4 - 1/\tau \| 1/2 + \varepsilon/2) \geq \tau(\operatorname{KL}(i/\tau \| 1/2 - \varepsilon/2) + \operatorname{KL}(j/\tau \| 1/2 - \varepsilon/2))$$
$$\geq \log(1/2\delta) - 2 - \log(2\pi) - \log(\tau) \ .$$

Hence using lemma F.5 and for $\tau > 2/\varepsilon$

$$2\tau \operatorname{KL}(1/2 + \varepsilon/4 \| 1/2 + \varepsilon/2) \geq -2\tau(\operatorname{KL}(1/2 + \varepsilon/4 - 1/\tau \| 1/2 + \varepsilon/2) - \operatorname{KL}(1/2 + \varepsilon/4 \| 1/2 + \varepsilon/2))$$
$$+ \log(1/2\delta) - 2 - \log(2\pi) - \log(\tau)$$
$$\geq -2\tau \int_{1/2+\varepsilon/4-1/\tau}^{1/2+\varepsilon/4} du \int_u^{1/2+\varepsilon/2} dv \frac{1}{v(1-v)} + \log(1/2\delta)$$
$$- 2 - \log(2\pi) - \log(\tau)$$
$$\geq -2(\varepsilon/4 + 1/\tau) \sup_{[1/2+\varepsilon/4-1/\tau, 1/2+\varepsilon/2]} \frac{1}{v(1-v)} + \log(1/2\delta)$$
$$- 2 - \log(2\pi) - \log(\tau)$$
$$\geq -2\varepsilon \sup_{[1/2, 1/2+\varepsilon]} \frac{1}{v(1-v)} + \log(1/2\delta) - 2 - \log(2\pi) - \log(\tau) \ .$$

Then lemma F.7 implies

$$\tau \geq \frac{-2\varepsilon \sup_{[1/2, 1/2+\varepsilon]} \frac{1}{v(1-v)} + \log(1/2\delta) - 2 - \log(2\pi)}{2 \operatorname{KL}(1/2 + \varepsilon/4 \| 1/2 + \varepsilon/2)} - \frac{\log\left( \frac{-2\varepsilon \sup_{[1/2, 1/2+\varepsilon]} \frac{1}{v(1-v)} + \log(1/2\delta) - 2 - \log(2\pi)}{2 \operatorname{KL}(1/2+\varepsilon/4 \| 1/2+\varepsilon/2)} \right)}{4 \operatorname{KL}(1/2 + \varepsilon/4 \| 1/2 + \varepsilon/2)}$$
$$\geq \frac{\log(1/2\delta)}{2 \operatorname{KL}(1/2 + \varepsilon/4 \| 1/2 + \varepsilon/2)} - \mathcal{O}\left( \frac{\log\log(1/\delta)}{\operatorname{KL}(1/2 + \varepsilon/4 \| 1/2 + \varepsilon/2)} \right) \ .$$

Finally we get the asymptotic lower bound:

$$\liminf_{\delta \to 0} \frac{\tau}{\log(1/\delta)} \geq \frac{1}{2 \operatorname{KL}(1/2 - \varepsilon/4 \| 1/2 - \varepsilon/2)}.$$

- If $C(i,j) = 1$, let $x$ and $y$ two words of length $\tau$ constituted of i.i.d samples from $\{1/2, 1/2, 0, \ldots, 0\}$, then $\mathbb{P}_{1/2,1/2}(N_1(x) = i, N_1(y) = j) \leq 2\delta$ hence with Stirling's approximation

$$\frac{e^{-2}}{2\pi\tau} e^{-\tau \,\mathrm{KL}(i/\tau\|1/2)} e^{-\tau \,\mathrm{KL}(1-j/\tau\|1/2)} \leq 2\delta \;.$$

Using the same lemmas as before, we get the following lower bound

$$\tau \geq \frac{\log(1/2\delta)}{2\,\mathrm{KL}(1/2 + \varepsilon/4\|1/2)} - \mathcal{O}\left(\frac{\log\log(1/\delta)}{\mathrm{KL}(1/2 + \varepsilon/4\|1/2)}\right).$$

Finally we get the asymptotic lower bound:

$$\liminf_{\delta \to 0} \frac{\tau}{\log(1/\delta)} \geq \frac{1}{2\,\mathrm{KL}(1/2 + \varepsilon/4\|1/2)} \;.$$

## A.2 Proof of Proposition 3.2

We propose the following general lower bounds for testing closeness.

**Lemma A.1.** *Let $T$ be a stopping rule for testing $\mathcal{D}_1 = \mathcal{D}_2$ vs $\mathrm{TV}(\mathcal{D}_1, \mathcal{D}_2) > \varepsilon$ with an error probability $\delta$. Let $\tau_1$ and $\tau_2$ the associated stopping times. We have*

- $\mathbb{E}(\tau_1(T, \mathcal{D}_1, \mathcal{D}_2)) \geq \frac{\log 1/3\delta}{\inf_{\mathcal{D}'_{1,2}\text{s.t. }\mathrm{TV}(\mathcal{D}'_1,\mathcal{D}'_2)>\varepsilon}\mathrm{KL}(\mathcal{D}_1\|\mathcal{D}'_1)+\mathrm{KL}(\mathcal{D}_2\|\mathcal{D}'_2)}$ *if $\mathcal{D}_1 = \mathcal{D}_2$.*
- $\mathbb{E}(\tau_2(T, \mathcal{D}_1, \mathcal{D}_2)) \geq \frac{\log 1/3\delta}{\inf_{\mathcal{D}}\mathrm{KL}(\mathcal{D}_1\|\mathcal{D})+\mathrm{KL}(\mathcal{D}_2\|\mathcal{D})}$ *if $\mathrm{TV}(\mathcal{D}_1, \mathcal{D}_2) > \varepsilon$.*

*Proof.* Similarly as in the previous proof, we consider the two different cases $\mathcal{D}' = \mathcal{D}$ and $\mathrm{TV}(\mathcal{D}', \mathcal{D}) > \varepsilon$.

**The case $\mathcal{D}_1 = \mathcal{D}_2$.** We denote by $\mathbb{P}_{\mathcal{D}_1, \mathcal{D}_2}$ the probability distribution on $([n] \times [n])^{\mathbb{N}}$ with independent marginals $(X_i, Y_i)$ of distribution $\mathcal{D}_1 \otimes \mathcal{D}_2$. Let $Z = (X_1, Y_1 \ldots, X_{\tau_1}, Y_{\tau_1})$. Let $\mathcal{D}'_1, \mathcal{D}'_2$ be two distributions such that $\mathrm{TV}(\mathcal{D}'_1, \mathcal{D}'_2) > \varepsilon$. Data processing property of Kullback-Leibler's divergence implies

$$\mathrm{KL}\left(\mathbb{P}^Z_{\mathcal{D}_1,\mathcal{D}_2}\|\mathbb{P}^Z_{\mathcal{D}'_1,\mathcal{D}'_2}\right) \geq \mathrm{KL}\left(\mathbb{P}_{\mathcal{D}_1,\mathcal{D}_2}(\tau_1 < \infty)\|\mathbb{P}_{\mathcal{D}'_1,\mathcal{D}'_2}(\tau_1 < \infty)\right) \;. \qquad (3)$$

By definition of $\tau_1$ we have $\mathbb{P}_{\mathcal{D}_1,\mathcal{D}_2}(\tau_1 < \infty) \geq 1 - \delta$ and $\mathbb{P}_{\mathcal{D}'_1,\mathcal{D}'_2}(\tau_1 < \infty) \leq \delta$. Tensorization property and Wald's lemma (F.4) lead to

$$\mathrm{KL}\left(\mathbb{P}^Z_{\mathcal{D}_1,\mathcal{D}_2}\|\mathbb{P}^Z_{\mathcal{D}'_1,\mathcal{D}'_2}\right) = \mathbb{E}(\tau_1(T,\mathcal{D}_1,\mathcal{D}_1))\,\mathrm{KL}(\mathcal{D}_1\|\mathcal{D}'_1) + \mathbb{E}(\tau_1(T,\mathcal{D}_1,\mathcal{D}_2))\,\mathrm{KL}(\mathcal{D}_2\|\mathcal{D}'_2) \;.$$

The inequality 3 becomes

$$\mathbb{E}(\tau_1(T,\mathcal{D}_1,\mathcal{D}_2))\,\mathrm{KL}(\mathcal{D}_1\|\mathcal{D}'_1) + \mathbb{E}(\tau_1(T,\mathcal{D}_1,\mathcal{D}_2))\,\mathrm{KL}(\mathcal{D}_2\|\mathcal{D}'_2) \geq \mathrm{KL}(1-\delta\|\delta) \geq \log 1/3\delta \;,$$

which is valid for all distribution $\mathcal{D}'_1$ and $\mathcal{D}'_2$ such that $\mathrm{TV}(\mathcal{D}'_1, \mathcal{D}'_2) > \varepsilon$, consequently

$$\mathbb{E}(\tau_1(T,\mathcal{D}_1,\mathcal{D}_2)) \geq \frac{\log 1/3\delta}{\inf_{\mathcal{D}'_{1,2}\text{s.t. }\mathrm{TV}(\mathcal{D}'_1,\mathcal{D}'_2)>\varepsilon}\mathrm{KL}(\mathcal{D}_1\|\mathcal{D}'_1) + \mathrm{KL}(\mathcal{D}_2\|\mathcal{D}'_2)} \;.$$

**The case $\mathrm{TV}(\mathcal{D}_1, \mathcal{D}_2) > \varepsilon$.** Likewise we prove for $Z = (X_1, Y_1 \ldots, X_{\tau_2}, Y_{\tau_2})$ and $\mathcal{D}$ a distribution on $[n]$.

$$\begin{aligned}
\mathbb{E}(\tau_2(T,\mathcal{D}_1,\mathcal{D}_2))\,\mathrm{KL}(\mathcal{D}_1\|\mathcal{D}) + \mathbb{E}(\tau_2(T,\mathcal{D}_1,\mathcal{D}_2))\,\mathrm{KL}(\mathcal{D}_2\|\mathcal{D}) &= \mathrm{KL}\left(\mathbb{P}^Z_{\mathcal{D}_1,\mathcal{D}_2}\|\mathbb{P}^Z_{\mathcal{D},\mathcal{D}}\right) \\
&\geq \mathrm{KL}\left(\mathbb{P}_{\mathcal{D}_1,\mathcal{D}_2}(\tau_2 < \infty)\|\mathbb{P}_{\mathcal{D},\mathcal{D}}(\tau_2 < \infty)\right) \\
&\geq \mathrm{KL}(1-\delta\|\delta) \\
&\geq \log 1/3\delta \;.
\end{aligned}$$

which is valid for all distribution $\mathcal{D}$, consequently

$$\mathbb{E}(\tau_2(T, \mathcal{D}_1, \mathcal{D}_2)) \geq \frac{\log 1/3\delta}{\inf_{\mathcal{D}} \text{KL}(\mathcal{D}_1 \| \mathcal{D}) + \text{KL}(\mathcal{D}_2 \| \mathcal{D})}.$$

$\square$

The proof of Proposition 3.2 follows from this Lemma by choosing for the first point $\mathcal{D}_1 = \mathcal{D}_2 = \{1/2, 1/2, 0, \ldots, 0\}$ and $\mathcal{D}'_{1,2} = \{1/2 \pm \varepsilon/2, 1/2 \mp \varepsilon/2, 0, \ldots, 0\}$. For the second point, we use $\mathcal{D} = \{1/2, 1/2, 0, \ldots, 0\}$ and $\mathcal{D}_{1,2} = \{1/2 \pm d/2, 1/2 \mp d/2, 0, \ldots, 0\}$.

## B    Analysis of Alg. 1

**Correctness of Alg. 1.**    We should prove that the Alg. 1 has an error probability less than $\delta$. We use the following lemma which can be proven using McDiarmid's inequality and union bounds.

**Lemma B.1.** *If $\{A_1, \ldots, A_t\}$ (resp $\{B_1, \ldots, B_t\}$ ) i.i.d. with the law $\mathcal{D}_1$ (resp $\mathcal{D}_2$), we have the following inequality*

$$\mathbb{P}\left( \exists t \geq 1, \exists B \subset [n/2] : \left| \tilde{\mathcal{D}}_{1,t}(B) - \mathcal{D}_1(B) - \tilde{\mathcal{D}}_{2,t}(B) + \mathcal{D}_2(B) \right| > \sqrt{\log\left( \frac{2^{n-1}t(t+1)}{\delta} \right) / t} \right) \leq \delta.$$

Using this lemma we can conclude:

- If $\mathcal{D}_1 = \mathcal{D}_2$, the probability of error is given by

$$\mathbb{P}\left( \tau_2 \leq \tau_1 \right) \leq \mathbb{P}\left( \exists t \geq 1 : \text{TV}\left( \tilde{\mathcal{D}}_{1,t}, \tilde{\mathcal{D}}_{2,t} \right) > \sqrt{\log\left( \frac{2^{n-1}t(t+1)}{\delta} \right) / t} \right) \leq \delta.$$

- If $\text{TV}(\mathcal{D}_1, \mathcal{D}_2) = |\mathcal{D}_1(B_{opt}) - \mathcal{D}_2(B_{opt})| > \varepsilon$, the probability of error is given by

$$\mathbb{P}\left( \tau_1 \leq \tau_2 \right) \leq \mathbb{P}\left( \exists t \geq 1 : \text{TV}\left( \tilde{\mathcal{D}}_{1,t}, \tilde{\mathcal{D}}_{2,t} \right) \leq \varepsilon - \sqrt{\log\left( \frac{2^{n-1}t(t+1)}{\delta} \right) / t} \right)$$

$$\leq \mathbb{P}\left( \exists t \geq 1 : \left| \tilde{\mathcal{D}}_{1,t}(B_{opt}) - \tilde{\mathcal{D}}_{2,t}(B_{opt})) \right| \leq \varepsilon - \sqrt{\log\left( \frac{2^{n-1}t(t+1)}{\delta} \right) / t} \right)$$

$$\leq \mathbb{P}\left( \exists t \geq 1 : \left| \tilde{\mathcal{D}}_{1,t}(B_{opt}) - \mathcal{D}_1(B_{opt}) - \tilde{\mathcal{D}}_{2,t}(B_{opt}) + \mathcal{D}_2(B_{opt})) \right| \geq |\mathcal{D}_1(B_{opt}) - \mathcal{D}_2(B_{opt})| \right.$$

$$\left. - \varepsilon + \sqrt{\log\left( \frac{2^{n-1}t(t+1)}{\delta} \right) / t} \right)$$

$$\leq \mathbb{P}\left( \exists t \geq 1 : \left| \tilde{\mathcal{D}}_{1,t}(B_{opt}) - \mathcal{D}_1(B_{opt}) - \tilde{\mathcal{D}}_{2,t}(B_{opt}) + \mathcal{D}_2(B_{opt})) \right| \right.$$

$$\left. > \sqrt{\log\left( \frac{2^{n-1}t(t+1)}{\delta} \right) / t} \right)$$

$$\leq \delta.$$

These computations prove the correctness of Alg. 1.

**Complexity of Alg. 1.** We study here the complexity of Alg. 1. To this aim, we make a case study and use lemma B.2 to upper bound the stopping rules.

**Lemma B.2.** *$T$ a random variable taking values in $\mathbb{N}$, we have for all $N \in \mathbb{N}^*$*

$$\mathbb{E}(T) \leq N + \sum_{t \geq N} \mathbb{P}(T \geq t) .$$

Let us take $\alpha \in (0,1)$,

- If $\mathcal{D}_1 = \mathcal{D}_2$, we take $N = \left\lceil \frac{\log(2^{n+1}/\delta)}{(\alpha\varepsilon)^2} \right\rceil + 1$ and $\tilde{\alpha} \in (0,1)$[1] so that

$$\tilde{\alpha}^2 = \alpha^2 \left( \frac{\log\log(2^{n+1}/\delta) - \log((\alpha\varepsilon)^2)}{\log(2^{n+1}/\delta)} + 1 \right).$$

The estimated stopping time can be bound as

$$\mathbb{E}(\tau_1(\mathcal{D}_1, \mathcal{D}_2)) \leq N + \sum_{s \geq N} \mathbb{P}(\tau_1(\mathcal{D}_1, \mathcal{D}_2) \geq s)$$

$$\leq N + \sum_{t \geq N-1} \mathbb{P}\left( \text{TV}\left(\tilde{\mathcal{D}}_{1,t}, \tilde{\mathcal{D}}_{2,t}\right) > \varepsilon - \sqrt{\log\left(\frac{2^{n-1}t(t+1)}{\delta}\right)/t} \right)$$

$$\leq N + \sum_{t \geq N-1} \mathbb{P}\left( \text{TV}\left(\tilde{\mathcal{D}}_{1,t}, \tilde{\mathcal{D}}_{2,t}\right) > \varepsilon - \tilde{\alpha}\varepsilon \right)$$

$$\leq N + \sum_{t \geq N-1} \mathbb{P}\left( \text{TV}\left(\tilde{\mathcal{D}}_{1,t}, \tilde{\mathcal{D}}_{2,t}\right) > (1 - \tilde{\alpha})\varepsilon \right)$$

$$\leq N + \sum_{t \geq N-1} 2^{n/2} e^{-t((1-\tilde{\alpha})\varepsilon)^2}, \text{(McDiarmid's inequality)}$$

$$\leq N + \frac{2^{n/2} e^{-(N-1)((1-\tilde{\alpha})\varepsilon)^2}}{1 - e^{-((1-\tilde{\alpha})\varepsilon)^2}}$$

$$\leq \frac{\log(2^{n+1}/\delta)}{(\alpha\varepsilon)^2} + 2\frac{2^{n/2} e^{-(N-1)((1-\tilde{\alpha})\varepsilon)^2}}{((1-\tilde{\alpha})\varepsilon)^2} + 1 \, , (1 - e^{-x} \geq x/2 \text{ for } 0 < x < 1)$$

$$\leq \frac{\log(2^{n+1}/\delta)}{\varepsilon^2} + \frac{\log(2^{n+1}/\delta)^{2/3}}{\varepsilon^2} + \mathcal{O}\left( \frac{\log(2^{n+1}/\delta)^{2/3}}{\varepsilon^2} \right)$$

$$\leq \frac{\log(2^{n+1}/\delta)}{\varepsilon^2} + \mathcal{O}\left( \frac{\log(2^{n+1}/\delta)^{2/3}}{\varepsilon^2} \right) ,$$

for $\alpha = (1 + \log(2^{n+1}/\delta)^{-1/3})^{-2}$ so that $1 - \tilde{\alpha} \geq C\log(2^{n+1}/\delta)^{-1/3}$ and we suppose here that $n < 2C^2 \log(2^{n+1}/\delta)^{1/3}$.

- If $d = \text{TV}(\mathcal{D}_1, \mathcal{D}_2) = |\mathcal{D}_1(B_{opt}) - \mathcal{D}_2(B_{opt})| > \varepsilon$, we take $N = \left\lceil \frac{\log(2^{n+1}/\delta)}{(\alpha d)^2} \right\rceil + 1$. We take $\tilde{\alpha} \in (0,1)$ so that $\tilde{\alpha}^2 = \alpha^2 \left( \frac{\log\log(2^{n+1}/\delta) - \log((\alpha d)^2)}{\log(2^{n+1}/\delta)} + 1 \right)$. The estimated stopping time can be

---

[1] for fixed $\alpha$ we take $\delta$ small enough to have $\tilde{\alpha} < 1$.

bound as

$$\mathbb{E}(\tau_2(\mathcal{D}_1, \mathcal{D}_2)) \leq N + \sum_{s \geq N} \mathbb{P}(\tau_2(\mathcal{D}_1, \mathcal{D}_2) \geq s)$$

$$\leq N + \sum_{t \geq N-1} \mathbb{P}\left(\mathrm{TV}\left(\tilde{\mathcal{D}}_{1,t}, \tilde{\mathcal{D}}_{2,t}\right) \leq \sqrt{\log\left(\frac{2^{n-1}t(t+1)}{\delta}\right)/t}\right)$$

$$\leq N + \sum_{t \geq N-1} \mathbb{P}\left(\mathrm{TV}\left(\tilde{\mathcal{D}}_{1,t}, \tilde{\mathcal{D}}_{2,t}\right) \leq \sqrt{\log\left(\frac{2^{n-1}t(t+1)}{\delta}\right)/t}\right)$$

$$\leq N + \sum_{t \geq N-1} \mathbb{P}\left(\left|\tilde{\mathcal{D}}_{1,t}(B_{opt}) - \tilde{\mathcal{D}}_{2,t}(B_{opt}))\right| \leq \sqrt{\log\left(\frac{2^{n-1}t(t+1)}{\delta}\right)/t}\right)$$

$$\leq N + \sum_{t \geq N-1} \mathbb{P}\left(\left|\tilde{\mathcal{D}}_{1,t}(B_{opt}) - \mathcal{D}_1(B_{opt}) - \tilde{\mathcal{D}}_{2,t}(B_{opt}) + \mathcal{D}_2(B_{opt}))\right|\right.$$

$$\left.> |\mathcal{D}_1(B_{opt}) - \mathcal{D}_2(B_{opt})| - \sqrt{\log\left(\frac{2^{n-1}t(t+1)}{\delta}\right)/t}\right)$$

$$\leq N + \sum_{t \geq N-1} \mathbb{P}\left(\left|\tilde{\mathcal{D}}_{1,t}(B_{opt}) - \mathcal{D}_1(B_{opt}) - \tilde{\mathcal{D}}_{2,t}(B_{opt}) + \mathcal{D}_2(B_{opt}))\right| > (1-\tilde{\alpha})d\right)$$

$$\leq N + \sum_{t \geq N-1} e^{-t((1-\tilde{\alpha})d)^2}$$

$$\leq N + \frac{e^{-(N-1)((1-\tilde{\alpha})d)^2}}{1 - e^{-((1-\tilde{\alpha})d)^2}}$$

$$\leq \frac{\log(2^{n+1}/\delta)}{(\alpha d)^2} + \frac{2}{(1-\tilde{\alpha})^2 d^2} + 1$$

$$\leq \frac{\log(2^{n+1}/\delta)}{d^2} + \mathcal{O}\left(\frac{\log(2^{n+1}/\delta)^{2/3}}{d^2}\right),$$

where we choose $\alpha = (1 + \log(2^{n+1}/\delta)^{-1/3})^{-2}$ and we use the inequality $1 - e^{-x} \geq x/2$ for $0 < x < 1$ in the last line.

Finally, we can deduce the limit when $\mathcal{D}_1 = \mathcal{D}_2$:

$$\limsup_{\delta \to 0} \frac{\mathbb{E}(\tau_1(\mathcal{D}_1, \mathcal{D}_2))}{\log(1/\delta)} \leq \limsup_{\delta \to 0} \frac{\log(2^{n+1}/\delta)}{\log(1/\delta)\varepsilon^2} + \mathcal{O}\left(\frac{\log(2^{n+1}/\delta)^{2/3}}{\log(1/\delta)\varepsilon^2}\right)$$

$$\leq \frac{1}{\varepsilon^2},$$

and when $d = \mathrm{TV}(\mathcal{D}_1, \mathcal{D}_2) > \varepsilon$:

$$\limsup_{\delta \to 0} \frac{\mathbb{E}(\tau_2(\mathcal{D}_1, \mathcal{D}_2))}{\log(1/\delta)} \leq \limsup_{\delta \to 0} \frac{\log(2^{n+1}/\delta)}{\log(1/\delta)d^2} + \mathcal{O}\left(\frac{\log(2^{n+1}/\delta)^{2/3}}{\log(1/\delta)d^2}\right)$$

$$\leq \frac{1}{d^2}.$$

This concludes the proof of the complexity of Alg. 1.

## C    Proof of Theorem 4.4

We prove both cases at once, to do so let $d = \varepsilon \vee \mathrm{TV}(\mathcal{D}_1, \mathcal{D}_2)$, $\tau = \tau_1$ if $d = 0$ and $\tau = \tau_2$ if $d > \varepsilon$, we know that $\mathbb{E}(\tau) \le \sum_{s \le N_d} \mathbb{P}(\tau \ge s) + \sum_{s > N_d} \mathbb{P}(\tau \ge s) \le N_d + \sum_{s > N_d} \mathbb{P}(\tau \ge s)$ so it suffices to prove that $\sum_{s > N_d} \mathbb{P}(\tau \ge s) \le N_d$. By the definitions of $\tau_1$ and $\tau_2$, $\tau \ge s$ implies $|Z_{s-1} - \mathbb{E}(Z_{s-1})| > \Delta_{s-1} - \Psi_{s-1}$ but we have chosen $N_d$ so that if $t = s - 1 \ge N_d$, $\Delta_{s-1} - \Psi_{s-1} \ge \frac{C}{2} \min \left\{ (s-1)d, \frac{(s-1)^2 d^2}{n}, \frac{(s-1)^{3/2} d^2}{\sqrt{n}} \right\}$. This last claim follows from Lemma F.8 in App. F.5. Finally

$$\sum_{s > N_d} \mathbb{P}(\tau \ge s) \le \sum_{t \ge N_d} \mathbb{P}\left( |Z_t - \mathbb{E}(Z_t)| > \frac{C}{2} \min \left\{ td, \frac{t^2 d^2}{n}, \frac{t^{3/2} d^2}{\sqrt{n}} \right\} \right)$$

$$\overset{\text{(McDiarmid's inequality)}}{\le} \sum_{t \ge N_d - 1} e^{-\frac{C^2}{16} \min \left\{ td^2, \frac{t^3 d^4}{n^2}, \frac{t^2 d^4}{n} \right\}} \le N_d \ .$$

The last inequality is proven in App. F.5. Our claim follows.

## D    Proof of Theorem 4.5

We prove only the first statement, the others being similar. Suppose that such a stopping rule exists. Let $d > \varepsilon$ and $m = c \frac{\sqrt{n \log(1/3\delta)}}{d^2}$. Let $U_n$ the uniform distribution and $D$ a uniformly chosen distribution where $D_i = \frac{1 \pm 2d}{n}$ with probability $1/2$ each. With the work of Diakonikolas and Kane [2016] (Section 3), we can show that $\mathrm{KL}(D^{\otimes Poi(m)} \| U_n^{\otimes Poi(m)}) \le C \frac{m^2 d^4}{n}$ where $C$ is a constant. Therefore

$$\begin{aligned} \mathrm{KL}(D^{\otimes m} \| U_n^{\otimes m}) &= m \, \mathrm{KL}(D \| U_n) \\ &= \mathbb{E}(Poi(m)) \, \mathrm{KL}(D \| U_n) \\ &= \mathrm{KL}(D^{\otimes Poi(m)} \| U_n^{\otimes Poi(m)}) \quad \text{(Wald's lemma)} \\ &\le C \frac{m^2 d^4}{n} \ . \end{aligned}$$

But

$$\begin{aligned} \mathrm{KL}(D^{\otimes m} \| U_n^{\otimes m}) &\ge \mathrm{KL}(\mathbb{P}_D(\tau_2 \le m) \| \mathbb{P}_{U_n}(\tau_2 \le m)) \\ &\ge \mathrm{KL}(1 - \delta \| \delta) \\ &\ge \log(1/3\delta) \ , \end{aligned}$$

since $\mathbb{P}_D(\tau_2 \le m) \ge 1 - \delta$ and $\mathbb{P}_{U_n}(\tau_2 \le m) = \mathbb{P}_{U_n}(\tau_2 \le m, \tau_1 < \tau_2) + \mathbb{P}_{U_n}(\tau_2 \le m, \tau_1 \ge \tau_2) \le \delta$. Hence

$$C \frac{\left( c \frac{\sqrt{n \log(1/3\delta)}}{d^2} \right)^2 d^4}{n} \ge \log(1/3\delta) \ ,$$

which gives the contradiction if $c < 1/\sqrt{C}$.

## E    Proof of Theorem 4.7

We prove here Theorem 4.7. We use ideas similar to Karp and Kleinberg [2007]. We prove only the first statement, the others being similar. Let's start by a lemma:

**Lemma E.1.** *Let $X$ and $Y$ two random variables and $E$ some event verifying $\mathbb{P}_X(E) \geq 1/3$ and $\mathbb{P}_Y(E) < 1/3$, we have*

$$\mathrm{KL}(\mathbb{P}_X \| \mathbb{P}_Y) \geq -\frac{1}{3} \log(3\mathbb{P}_Y(E)) - \frac{1}{e}.$$

*Proof.* By data processing property of Kullback-Leibler's divergence:

$$\mathrm{KL}(\mathbb{P}_X \| \mathbb{P}_Y) \geq \mathrm{KL}(\mathbb{P}_X(E) \| \mathbb{P}_Y(E))$$
$$\geq \mathbb{P}_X(E) \log \frac{\mathbb{P}_X(E)}{\mathbb{P}_Y(E)} + (1 - \mathbb{P}_X(E)) \log \frac{1 - \mathbb{P}_X(E)}{1 - \mathbb{P}_Y(E)}$$
$$\geq -\frac{1}{3} \log(3\mathbb{P}_Y(E)) + (1 - \mathbb{P}_X(E)) \log(1 - \mathbb{P}_X(E))$$
$$\geq -\frac{1}{3} \log(3\mathbb{P}_Y(E)) - \frac{1}{e} .$$

$\square$

Suppose by contradiction that there is a stopping rule such that

$$\mathbb{P}\left(\tau_2(T, \mathcal{D}_1, \mathcal{D}_2) > \frac{n^{1/2} \log\log(1/d)^{1/2}}{Cd^2}\right) \leq \frac{1}{16} ,$$

whenever $d = \mathrm{TV}(\mathcal{D}_1, \mathcal{D}_2) > 0$. Let $\varepsilon_1 = 1/3$, we construct recursively $T_k = \left\lceil \frac{n^{1/2} \log\log(1/\varepsilon_k)^{1/2}}{C\varepsilon_k^2} \right\rceil = \frac{C'\sqrt{n}}{\varepsilon_{k+1}^2}$ where $C$ and $C'$ are constants defined later. For each integer $j$, we take $m_j \sim Poi(j)$. Let $U_n$ the uniform distribution and $D_k$ a uniformly chosen distribution where $D_{k,i} = \frac{1 \pm 2\varepsilon_k}{n}$ with probability $1/2$ each. With the work of Diakonikolas and Kane [2016] (Section 3), we can show that $\mathrm{KL}(U_n^{\otimes m_j} \otimes D_k^{\otimes m_j} \| U_n^{\otimes m_j} \otimes U_n^{\otimes m_j}) \leq C'' \frac{j^2 \varepsilon_k^4}{n}$ where $C''$ is a constant. Since $\mathrm{TV}(U_n, D_k) = \varepsilon_k > 0$, $\mathbb{P}(\tau_2(T, U_n, D_k) > T_k) \leq 1/16$. Let $E_k$ be the event that the stopping rule decides that the distributions are not equal between $T_{k-1}$ and $T_k$. We have $\mathbb{P}(\tau_2(T, U_n, D_k) \leq T_{k-1}) \leq 1/3$ since otherwise Lemma E.1 implies:

$$-\frac{1}{3} \log\left(3\mathbb{P}(\tau_2(T, U_n, U_n) \leq T_{k-1})\right) - \frac{1}{e} \leq \mathrm{KL}(U_n^{\otimes m_{T_{k-1}}} \otimes D_k^{\otimes m_{T_{k-1}}} \| U_n^{\otimes m_{T_{k-1}}} \otimes U_n^{\otimes m_{T_{k-1}}})$$
$$\leq C'' \frac{T_{k-1}^2 \varepsilon_k^4}{n}$$
$$\leq C''C' ,$$

thus

$$\mathbb{P}(\tau_2(T, U_n, U_n) \leq T_{k-1}) \geq e^{-3C''C' - 3/e}/3 > 0.1,$$

for good choice of $C'$ and this contradicts the fact the the stopping rule is infinite with a probability at least 0.9. The stopping rule is 0.1 correct so $\mathbb{P}(\tau_2(T, U_n, D_k) < +\infty) \geq 0.9$ then

$$\mathbb{P}(T_{k-1} < \tau_2(T, U_n, D_k) \leq T_k) \geq 0.9 - 1/3 - 1/16 > 0.5.$$

The same inequalities for the Kullback-Leibler's divergence as above permits to deduce:

$$1 \geq \sum_{k \geq 1} \mathbb{P}(T_{k-1} < \tau_2(T, U_n, U_n) \leq T_k) \geq \sum_{k \geq 1} \frac{1}{3} e^{-3C''T_k^2 \varepsilon_k^4/n - 3/e}$$
$$\geq \sum_{k \geq 1} \frac{1}{3e^2} e^{-3C''/C^2 \log\log(1/\varepsilon_k)} \text{ and choosing } C \text{ st } 3C''/C^2 = 1/2$$
$$\geq \sum_{k \geq 1} \frac{1}{3e^2} \frac{1}{\sqrt{\log(1/\varepsilon_k)}} .$$

But the later sum is divergent because if we denote $a_k = \log(1/\varepsilon_k)$, we have $a_{k+1} \leq a_k + \frac{1}{4} \log\log a_k + \mathcal{O}(1)$ thus $a_k = \mathcal{O}(k \log\log k)$ therefore $\frac{1}{\sqrt{\log(1/\varepsilon_k)}} \geq \frac{c}{k}$ which is divergent.

# F Technical lemmas

## F.1 Kullback-Leibler divergence

**Definition F.1** (Kullback Leibler divergence). *The Kullback Leibler divergence is defined for two distributions $p$ and $q$ on $[n]$ as*

$$\text{KL}(p\|q) = \sum_{i=1}^{n} p_i \log\left(\frac{p_i}{q_i}\right) .$$

*We denote by* $\text{KL}(p\|q) = \text{KL}(\mathcal{B}(p)\|\mathcal{B}(q))$.

Kullback-Leibler's divergence satisfies data-processing and tensorization properties:

**Proposition F.2.** *Let $p, p', q$ and $q'$ distributions on $[n]$, we have*

- ***Non negativity*** $\text{KL}(p\|q) \geq 0$.
- ***Data processing*** *Let $X$ a random variable and $g$ a function. Define the random variable $Y = g(X)$, we have*

$$\text{KL}\left(p^X\|q^X\right) \geq \text{KL}\left(p^Y\|q^Y\right). \tag{4}$$

- ***Tensorization***

$$\text{KL}(p \otimes p'\|q \otimes q') = \text{KL}(p\|q) + \text{KL}(p'\|q').$$

## F.2 Poissonization

The Poisson law of parameter $\lambda$ is denoted $Poi(\lambda)$ and defined as follows.

$$\forall k \in \mathbb{N}, \quad \mathbb{P}(Poi(\lambda) = k) = \frac{\lambda^k}{k!}e^{-\lambda} .$$

Poisson law is important for the analysis of testing' algorithms. In fact, some important random variables becomes independent when we take a number of samples following a Poisson law.

**Lemma F.3** (Poissonization). *Let $k \sim Poi(\tau)$ and $X = (X_1, \ldots, X_k)$ i.i.d samples from a distribution $p$ on $[n]$. For $i \in [n]$, we denote $Y_i$ the number of times $i$ appears in the tuple $X$. We have*

1. *$\{Y_1, \ldots, Y_n\}$ are independent.*

2. *For all $i \in [n]$, $Y_i \sim Poi(\tau p_i)$.*

## F.3 Wald's lemma

**Lemma F.4** (Wald [1944]). *Let $(X_n)_{n \geq 0}$ i.i.d random variables and $N \in \mathbb{N}$ a random variable independent of $(X_n)_n$. Suppose that $N$ and $X_1$ have finite expectations. we have*

$$\mathbb{E}(X_1 + \cdots + X_N) = \mathbb{E}(N)\mathbb{E}(X_1) .$$

### F.4 Modified McDiarmid's inequality

*Proof.* The proof uses similar arguments of Howard et al. [2018]. Actually $Z_t$ is a function of $4t$ variables (the samples from the distributions) and has the property $(2, \ldots, 2)$-bounded differences. McDiarmid's inequality implies $\mathbb{P}\left(\exists t \geq 1 : |Z_t - \mathbb{E}[Z_t]| \geq a + 4bt/a\right) \leq 2e^{-2b}$, taking the intervals $I_k = [\eta^k, \eta^{k+1})$ for $k$ integer we deduce for $b_k = \frac{1}{2}\log\left(\frac{2(k+1)^s}{\zeta(s)^{-1}\delta}\right)$ and $a_k = \frac{b_k}{a_k}\eta^{k+1}$ that

$$
\begin{aligned}
\mathbb{P}\left(\exists t \geq 1 : |Z_t - \mathbb{E}[Z_t]| \geq J(\eta, s, 4t)\right) &\leq \sum_{k \geq 0} \mathbb{P}\left(\exists t \in I_k : |Z_t - \mathbb{E}[Z_t]| \geq J(\eta, s, 4t)\right) \\
&\leq \sum_{k \geq 0} \mathbb{P}\left(\exists t \in I_k : |Z_t - \mathbb{E}[Z_t]| \geq a_k + 4b_k t/a_k\right) \\
&\leq \sum_{k \geq 0} 2e^{-2b_k} \leq \sum_{k \geq 0} \delta\frac{\zeta(s)^{-1}}{(k+1)^s} \leq \delta \ .
\end{aligned}
$$

$\square$

### F.5 Tools for non asymptotic inequalities

We group here different lemmas that help us to deal with the kl-divergence or logarithmic relations in order to find non asymptotic results. We start by giving some useful lemmas for the Kullback-Leibler's divergence between Bernoulli variables.

**Lemma F.5** (Lemmas for kl-divergence.). *Let $q > p$ two numbers in $[0, 1]$. Then*

- $2(p-q)^2 \leq \mathrm{KL}(p\|q) \leq \frac{(p-q)^2}{q(1-q)}$,
- $\mathrm{KL}(p\|q) \underset{q \to p}{\sim} \frac{(p-q)^2}{2q(1-q)}$,
- $\mathrm{KL}(q\|p) = \int_p^q du \int_p^u dv \frac{1}{v(1-v)}$.

**Sketch of proof.** The LHS of the first inequality is Pinsker's inequality, the RHS can be proven using the inequality $\log(1 + x) \leq x$, the second equivalence can be found by developing the $\log$ function and the third equality is proven by calculating the integral.

**Lemma F.6.** *[Developing kl]Let $q, \varepsilon$ and $\alpha$ positive real numbers such that $q + \varepsilon < 1$ and $\alpha < 1$, we have for $\alpha$ close enough to $1$*

$$
\frac{1}{\mathrm{KL}(q + \alpha\varepsilon\|q)} \leq \frac{1}{\mathrm{KL}(q + \varepsilon\|q)} + (1 - \alpha)\sup_{[q, q+\varepsilon]}\frac{1}{x(1-x)} \ .
$$

*Proof.* We use the inequality $\frac{1}{1-x} \leq 1 + 2x$ for $0 < x < 1/2$. We write

$$
\frac{1}{\mathrm{KL}(q + \alpha\varepsilon\|q)} = \frac{1}{\mathrm{KL}(q + \varepsilon\|q)(1 - x)} \ ,
$$

where $x = \frac{\text{KL}(q+\varepsilon\|q) - \text{KL}(q+\alpha\varepsilon\|q)}{\text{KL}(q+\varepsilon\|q)} < \frac{1}{2}$ if $\alpha$ is close enough to 1. Hence

$$
\begin{aligned}
\frac{1}{\text{KL}(q+\alpha\varepsilon\|q)} &\leq \frac{1}{\text{KL}(q+\varepsilon\|q)(1-x)} \\
&\leq \frac{1}{\text{KL}(q+\varepsilon\|q)}(1+2x) \\
&\leq \frac{1}{\text{KL}(q+\varepsilon\|q)} + 2\frac{\text{KL}(q+\varepsilon\|q) - \text{KL}(q+\alpha\varepsilon\|q)}{\text{KL}(q+\varepsilon\|q)^2} \\
&\leq \frac{1}{\text{KL}(q+\varepsilon\|q)} + \frac{2}{\text{KL}(q+\varepsilon\|q)^2} \int_{q+\alpha\varepsilon}^{q+\varepsilon} du \int_q^u dv \frac{1}{v(1-v)} \\
&\leq \frac{1}{\text{KL}(q+\varepsilon\|q)} + \frac{2(1-\alpha)\varepsilon^2}{\text{KL}(q+\varepsilon\|q)^2} \sup_{[q,q+\varepsilon]} \frac{1}{v(1-v)} \\
&\leq \frac{1}{\text{KL}(q+\varepsilon\|q)} + \frac{2(1-\alpha)\varepsilon^2}{2\varepsilon^2} \sup_{[q,q+\varepsilon]} \frac{1}{v(1-v)} \\
&\leq \frac{1}{\text{KL}(q+\varepsilon\|q)} + (1-\alpha) \sup_{[q,q+\varepsilon]} \frac{1}{v(1-v)} \, .
\end{aligned}
$$

$\square$

When we deal with inequalities involving $t$ and $\log t$ (or $\log\log t$) and want to deduce inequalities only on $t$, the following lemma proves to be useful.

**Lemma F.7.** *Let $t, a > 1$ and $b$ real numbers. We have the following implications:*

- *If $b \geq a + 1$ :*
$$ t \geq b + 2a \log(b) \Rightarrow t \geq b + a \log(t) \, , $$

- *If $b \geq 1$ :*
$$ t \geq b - a \log(t) \Rightarrow t \geq b - a \log(b) \, , $$

- *If $b \geq 2a$ :*
$$ t \geq b + 2a \log(\log(b) + 1) \Rightarrow t \geq b + a \log(\log(t) + 1) \, . $$

*Proof.* We prove only the first statement, the others being similar. Let $f(t) = t - b - a\log(t)$, we have $f'(t) = 1 - a/t$ thus $f$ is increasing on $(a, +\infty)$. Let $t \geq b + 2a\log(b) > a$,

$$
\begin{aligned}
f(t) \geq f(b + 2a\log(b)) &= b + 2a\log(b) - b - a\log(b + 2a\log(b)) \\
&= a\log(b) - a\log(1 + 2a\log(b)/b) \\
&\geq a\log(1+a) - a\log(1 + 2ab/eb) \quad \text{because } \log(b) \leq b/e \\
&\geq 0 \, .
\end{aligned}
$$

$\square$

For instance, by applying this lemma, we can obtain:

**Lemma F.8.** *Recall the definition of $N_\eta$:*

$$
\begin{aligned}
N_\eta = \max \Bigg\{ &\frac{128}{C^2} \frac{\log(\frac{\pi^2}{3\delta})}{\eta^2} + \frac{512e}{C^2\eta^2} \log\left(\log\left(\frac{128 \log(\frac{\pi^2}{3\delta})}{\eta^2 C^2}\right) + 1\right) + \frac{16c^2}{C^2\eta^2}, \\
&\left(\frac{128}{C^2} \frac{n^2 \log(\frac{\pi^2}{3\delta})}{\eta^4} + \frac{512en^2}{C^2\eta^4} \log\left(\log\left(\frac{128}{C^2} \frac{n^2 \log(\frac{\pi^2}{3\delta})}{\eta^4}\right) + 1\right) + \frac{16c^2n^2}{\eta^4 C^2}\right)^{1/3}, \\
&\left(\frac{128}{C^2} \frac{n \log(\frac{\pi^2}{3\delta})}{\eta^4} + \frac{512en}{C^2\eta^4} \log\left(\log\left(\frac{128}{C^2} \frac{n \log(\frac{\pi^2}{3\delta})}{\eta^4}\right) + 1\right) + \frac{16c^2n}{\eta^4 C^2}\right)^{1/2} \Bigg\} \, .
\end{aligned}
$$

*Let $\eta > 0$, if $t \geq N_\eta$, then*

$$\min\left\{t\eta, \frac{t^2\eta^2}{n}, \frac{t^{3/2}\eta^2}{\sqrt{n}}\right\} \geq \frac{4}{C}\sqrt{2t\log\left(\frac{\pi^2}{3\delta}\right) + 4et\log(\log(t)+1)} + \frac{2c}{C}\sqrt{t}.$$

Finally, the next lemma shows that the complexity of Alg. 2 cannot exceed $N_{d\vee\varepsilon}$ very much.

**Lemma F.9.** *We have for all $d > 0$:* $\sum_{t\geq N_d} e^{-\frac{C^2}{16}\min\left\{td^2, \frac{t^3d^4}{n^2}, \frac{t^2d^4}{n}\right\}} \leq N_d$.

*Proof.* We have

$$
\begin{aligned}
\sum_{t\geq N_d} e^{-\frac{C^2}{16}\min\left\{td^2, \frac{t^3d^4}{n^2}, \frac{t^2d^4}{n}\right\}} &\leq \sum_{t\geq nd^{-2}} e^{-\frac{C^2}{16}td^2} + \sum_{n\geq t\geq N_d-1} e^{-\frac{C^2}{16}\frac{t^3d^4}{n^2}} + \sum_{nd^{-2}>t>n} e^{-\frac{C^2}{16}\frac{t^2d^4}{n}} \\
&\leq \sum_{t\geq nd^{-2}} e^{-\frac{C^2}{16}td^2} + \sum_{n\geq t\geq N_d-1} e^{-2C^{1/3}\frac{td^{4/3}}{n^{2/3}}} + \sum_{nd^{-2}>t>n} e^{-\frac{C}{2}\frac{td^2}{\sqrt{n}}} \\
&\leq \frac{1}{1-e^{-\frac{C^2}{16}d^2}} + \frac{1}{1-e^{-2C^{1/3}\frac{d^{4/3}}{n^{2/3}}}} + \frac{1}{1-e^{-\frac{C}{2}\frac{d^2}{\sqrt{n}}}} \\
&\leq \frac{32}{C^2d^2} + \frac{n^{2/3}}{C^{1/3}d^{4/3}} + \frac{4\sqrt{n}}{Cd^2} \quad \text{since } 1-e^{-x} \geq x/2 \text{ for } 0 < x < 1 \\
&\leq N_d.
\end{aligned}
$$

$\square$

**Acknowledgement.**
Aurélien Garivier acknowledges the support of the Project IDEXLYON of the University of Lyon, in the framework of the Programme Investissements d'Avenir (ANR-16-IDEX-0005), and Chaire SeqALO (ANR-20-CHIA-0020).