# OpenReview forum: "Sequential Algorithms for Testing Closeness of Distributions"
_NeurIPS.cc/2021/Conference — NeurIPS 2021 Spotlight_

### Official Review · Reviewer_yo4x · 2021-07-14

**Rating:** 6
**Confidence:** 3

**Summary:**

In distribution property testing, one aims to solve decision problems on probability distributions with some slack between instances that should be accepted and instances that should be rejected (instances in the slack may be accepted or rejected). In this paper, the authors consider testing whether two discrete distributions with support size n are the same, or differ in total variation distance by at least some constant $\epsilon$. In particular, they compare the standard batch / non-adaptive setting, where an algorithm specififies a number of samples to draw from both distributions i.i.d. in advance, and the sequential setting, where an algorithm may query sample by sample until it decides it has seen enough. (i) Their sequential algorithm improves the asymptotical dependence of the best known and optimal batch algorithm's sample complexity from $\epsilon$ to $\max(\epsilon, TVD)$, where $TVD$ is the total variation distance of the two input distributions. In other words, given a lower bound on the total variation distance, the algorithms performs best possible in the sense it would not improve if a tight lower bound was known. The resulting algorithm has sample complexity roughly $O(n^{2/3} / \max(\epsilon, TVD) \cdot polylog(\delta))$. The authors show that their algorithm is asymptotically optimal. (ii) Furthermore, they show that if no lower bound is known, the algorithm stops after roughly $O(n^{2/3} / d^2 \cdot polylog(\log TVD))$ samples (provided $TVD > 0$), which improves the dependence on $n$ by roughly a factor $n^{1/3}$ compared to previous work and is shown to be optimal. (iii) For error probability $\delta$ and $n << \log(1 / \delta)$, they show that the query complexity of the best sequential algorithms is separated from the best batch algorithms by a constant factor of 4.

**Limitations And Societal Impact:**

Questions to authors:
1. I'm really confused what the asymptotical benefit of the sequential algorithm over the batch algorithm with an exponential-search like procedure for $\epsilon' \rightarrow \epsilon$ is, for both $\epsilon > 0$ and $\epsilon = 0$. The former case seems comparable to me, while the latter has a $\log(\log(1/d))$ dependence when the batch approach might have a $\log(1/d)$ dependence (which results from choosing an exponentially decreasing error probability). Could you confirm whether this is the case, and if not, why? To be precise, I'm thinking of this algorithm:
batch_to_sequential($\epsilon$)
  $\epsilon' = 1$
  result = true
  while result==true and $\epsilon < \epsilon'$:
    $\epsilon' = \epsilon' / 2$
    result = result && batch_algorithm($\epsilon'$) //accept = true
  return result

2. I'm not sure about how I should read the factor 4. To me, it sums up to "asymptotically, sequential algorithms have no advantage over batch algorithms; however, their absolute complexities are still separated by a constant factor". Maybe you could carve out and stress your interpretation?

**Main Review:**

Sequential algorithms have a practical advantage over batch algorithms: their complexity may adapt to the real edit distance / TVD instead of the lower bound $\epsilon$. This way, one can specify a small $\epsilon$ and still benefit from a large TVD of the actual distributions when it comes to sample complexity. A standard approach to simulate this using batch algorithms would be to start with a large $\epsilon'$, say $\epsilon'=1$, and then iterate by decreasing $\epsilon'$ each time by some factor, say $1/2$, and restarting until $\epsilon = \epsilon'$ or the algorithm rejects. As far as I can judge at a glance, it seems to me that this results in asymptotically comparable bounds at least for $\epsilon > 0$. In summary, it is not very clear to me how strong the improvement of the theoretical analysis of case (i), and to some extend, also of case (ii), is. As of now, I can see that the dependence $\log(\log(TVD))$ improves over $\log(TVD)$, which I think results from using a batch algorithm as described before, in case (ii), and the separating constant factor 4 of the tight bounds in case (iii). From a theoretical perspective, this would imply a negative result regarding the advantages of sequential algorithms for this problem in the case of $\epsilon > 0$ and a minor improvement for the case $\epsilon = 0$. For case (i), I don't see why the standard approach for batch algorithms doesn't yield similar bounds. It would be interesting to see how the sequential algorithm performs against the actual batch algorithm in practice.

**Time Spent Reviewing:**

4

---

> ### Author Response · Authors · 2021-08-10
> **Comment to the review of Reviewer yo4x**
>
> Thank you for your very insightful review.
>
> 1- Yes, we could use the batch algorithm in the way described in the review. The complexity is then roughly (there is a maximum of three terms, we kept only one term for simplicity) $O(\log(1/TVD)n^{2/3}\log(\log(1/TVD)/\delta)^{1/3} / (\max(\epsilon, TVD))^{4/3})$, and this complexity is worse than the complexity given in Theorem 4.4: the factor $\log(1/TVD)$ is then replaced by a ($\log\log(1/TVD))^{1/3}$, and we want to highlight that the exponents are also important. Furthermore, we preferred this bound rather than the algorithm proposed by the reviewer because it allows us to deduce directly the upper bound on the complexity of testing different in Theorem 4.6 ($\epsilon=0$) which is optimal up to constants.
> Nevertheless, we agree that this is an interesting suggestion to discuss in the paper, which we will do in the final version.
>
> 2- For large $n$, our results show that we cannot hope to improve batch algorithms more than eventually a constant and replacing $\epsilon$ by $\max(\epsilon, TV)$. However for small $n$, the (leading terms of the) sample complexities in batch and sequential settings are fully characterised up to a factor 1, and this shows that indeed there is a difference of a factor 4 between the two settings. This ratio 4 in the sample complexity is obviously a very significant gain in many applications.

---

> > ### Comment · Reviewer_yo4x · 2021-08-24
> > **Ack**
> >
> > Thank you for your response and clarifications! It eased my concerns, so that I increased my score by 1.

---

> > ### Public Comment · ~Clément_L_Canonne1 · 2021-11-19
> > **Doubling search**
> >
> > I am quite confusing by the answer to (1) here. A doubling search with parameters
> > $\varepsilon_j = 1/2^j$
> > and probability of failure
> > $\delta_j = \frac{\delta}{2j^2}$
> > in conjunction with the existing "batch" algorithms from closeness testing *will* give the right bounds (those claimed in this paper) including the loglog term, and the proof is very simple (basically, all you need is to set $\delta_j$ to form a convergent series, use a union bound, and analyze the sample complexity by a simple summation -- the rest is a  blackbox use of known testers).
> >
> > The statement in (1) is also not correct as stated, as it hides that loglog factor (which *is* in the theorem proven later, see Theorem 4.4).
> >
> > Finally, the abstract is a bit misleading, as the cited ESA'17 work does not lead to $n/\log n$ for closeness testing, but gives this for *olerant* closeness testing (a strictly harder question, for which that dependence on the domain size is necessary).
> >
> > Overall, I wonder how this set of reviews (one of the reviewers explicitly mentioning the double search, which, again, appears to easily give the upper bounds claimed) along with this set of scores led to this outcome. The authors did also not end up mentioning this approach at all, despite the response to the reviewer...

---

> > > ### Public Comment · ~Clément_L_Canonne1 · 2021-11-20
> > > **(typos)**
> > >
> > > confusing $\leadsto$ confused
> > >
> > > olerant $\leadsto$ tolerant
> > >
> > > Please let me know if I am missing an important point here...

---

> > > ### Public Comment · Authors · 2021-12-03
> > > **Doubling search**
> > >
> > > Thank you for your comment. Contrary to what we responded in the rebuttal, we now understand that the doubling search with the indicated parameters does give the same order of sample complexity as Theorem 4.4, and deserves to be cited. However, describing and commenting this alternative method requires a sufficient amount of space. In particular, it requires to explain why it does not contradict the lower bound on tolerant testing— $\Omega(n/\log(n))$ since we cannot ensure for different distributions of TV distance less than $\epsilon$ that the proposed algorithm will respond the right answer (the black box algorithm can return any hypothesis $H_0$ or $H_1$ if the TV distance is strictly between $0$ and $\epsilon$) hence the doubling search algorithm (as described) cannot be used for tolerant testing.  Moreover, in all experiments the actual sample complexity of our proposed algorithm appears to be better by an important constant factor.  This does not show off in the bounds (since the multiplicative constants are not known), but this can be understood at least when the TV distance $d$ satisfies $\epsilon< 2^{-k-1}< d = 2^{-k}(1-\eta)  < 2^{-k}$, when the doubling search obviously requires $\approx 4$ times more samples than necessary. Actually, even without this discretization effect, the difference is significant. This sub-optimality of the doubling search algorithm can be seen clearly for small alphabets where we can characterize the sample complexity to the constant: We gain a factor $4$ using our approach while using doubling search algorithm requires up to $4$ times more than the batch sample complexity when the $TV$ distance is strictly greater than $\epsilon$. As said before, limited by the lack of space, we were not able to include this discussion in the camera ready paper version. We rather chose to follow the AC recommendation regarding the references in the related work. We have added this discussion (which we definitely consider important and interesting) into the Arxiv version of the paper, which we plan to submit by the end of the year.
> > >
> > > In the ESA'17 paper, it is clearly stated that we don't ask the algorithm to stop when $D_1=D_2$ and want to minimize the sample complexity when $D_1\neq D_2$. For this specific problem, we have shown that $n/\log(n)$ samples  are not necessary and this can be done instead with the complexity given in Theorem 4.6. Furthermore, we gave a matching lower bound—Theorem 4.7.

---

> > > > ### Public Comment · ~Clément_L_Canonne1 · 2021-12-10
> > > > **Tolerant testing remark**
> > > >
> > > > Thanks for your comment. I cannot really comment on the small-alphabet part, but regarding your comment on the tolerant testing aspect: I don't quite see what you mean (as far as I can tell, there is no paradox or issue to resolve?)
> > > > - In the equality case ($p=q$), early rejection is not an issue as $\varepsilon=0$;
> > > > - in the non-equality case: if the search accepts for a larger $\varepsilon$ it's fine, as it will eventually reject once it reaches the right value (that's what the doubling search is about: you'll reject later); while if the blackbox algorithm rejects early (for larger $\varepsilon$, that's actually good, since it means it gave the right answer using fewer samples.

---

### Official Review · Reviewer_fYML · 2021-07-17

**Rating:** 6
**Confidence:** 3

**Summary:**

In this paper, the authors look at the problem of the goodness of fit testing in the sequential model. There has been a plethora of results and techniques in the last 20 years for the batch model. Here the authors allow the algorithm to stop at any point in time or ask for more samples. They show that the sequential setting beats the batch setting by at least a factor of 4 in the sample complexity. In fact, for a Boolean sample space, they have a more precise characterization up to the lower order terms in the batch setting. For the n>2 cases, they improve the dependence on the farness parameter from $\epsilon$ to $max(\epsilon, TV(D_1,D_2))$, which in particular improves upon a result by Daskalakis and Kawase for the case when $epsilon=0$.

**Limitations And Societal Impact:**

Yes

**Main Review:**

I think the findings are interesting, but the techniques are somewhat limited and incremental, as mentioned below. Therefore, I reckon the paper is marginally above acceptance threshold.
- Some results in this paper are claimed tight up to multiplicative constants, which is standard in property testing. Then the headline result: that sequential algorithms beat batch algorithm by a factor 4, for n=constant, looks rather weak to me. Although I understand that mathematics has a finer sense of tightness.
- Lines 69-73: actually, testing in chi-squared or KL is well-known to be in general impossible. See this paper titled: `which distribution distances are sub-linearly testable’.
- Section 4.1 (Algorithm 2): the techniques are from Diakonikolas et al. and I found their tester is not telling anything new technically.


**Time Spent Reviewing:**

10

---

> ### Author Response · Authors · 2021-08-10
> **Comment to the review of Reviewer fYML**
>
> Thank you for your very insightful review.
> We are convinced that gaining a factor $4$ in sample complexity may have a considerable impact, especially when the samples are costly or difficult to produce -- in fact, many statistics contributions are considered relevant even with much more modest gains.

---

### Official Review · Reviewer_fx8n · 2021-07-17

**Rating:** 6
**Confidence:** 4

**Summary:**

The authors tackle (a version of) the two-sample testing problem, where the distributions in question are known to be discrete.  This is clearly a fundamental problem across both theoretical and applied machine learning / statistics.

Letting $\mathcal D_1, \mathcal D_2$ be two distributions on $\{1,\ldots,n\}$, and $\varepsilon > 0$ be some fixed tolerance level, the authors consider testing:
$$
H_0: \mathcal D_1 = \mathcal D_2 \quad \textrm{vs.} \quad H_1: \textrm{TV}(\mathcal D_1, \mathcal D_2) > \varepsilon,
$$

where $\textrm{TV}$ denotes the usual total variation operator.

The authors study the above problem in both (i) the batch setting, where $\tau > 0$ samples from both $\mathcal D_1, \mathcal D_2$ are available, a priori; and (ii) the sequential setting, where (pairs of) draws from the two distributions stream in, one at a time.  The authors give minimax lower and upper bounds on the testing risk (phrased in the paper in terms of sample complexity, i.e., in terms of $\tau$); actually, the authors give pairs of bounds (one pair for each problem setting): one bound for $n \geq 2$, and another for $n \geq 3$.  A single simulation study is presented.

**Limitations And Societal Impact:**

Aside from my comments re: quality above, I'm not sure this section really applies here.

**Main Review:**

Overall, the authors tackle an important problem.  And the paper (being a resubmission from COLT) appears to be very solid.  My main issue is that the paper does not really situate itself properly in the literature.  I give some detailed feedback below, broken down into sections according to the categories suggested by the NeurIPS reviewer guidelines.

**Originality.**  I suppose this is a good place to expand on my comment re: related work that I made above.  Two-sample testing is of course a fundamental problem in statistics, and has a long history there.  So, I'm not entirely sure why the authors essentially disregard that entire literature, and only focus on citing TCS papers, in their paragraph on "Discussion on the setting and related work" (line 65).  The Kolmogorov-Smirnov and chi-squared examples of relevant tests, though there are others.  And as far as sequential testing goes, the authors might want to give this paper (+ the references therein plus follow-up work) a look: https://arxiv.org/pdf/1506.03486.pdf.

It's true that much of the above mentioned work concentrates on the situation where the underlying distributions are continuous, but there are remedies for that.  It's also true that these works focus on the K-S distance, though this object is certainly related to the TV distance; in any event, this issue raises the question of why we should care about testing for closeness in the TV sense in the first place (which the authors essentially say comes down to mathematical convenience ~line 66).

At the very minimum, I think the above works (and more) should be cited; probably more warranted would be a discussion of how the methods and theory in these two literatures compare.

**Clarity.**  The paper is pretty well written, though I found a number of typos:
* Lines 25-27: I don't understand this.  What you are doing is no different than hypothesis testing, where you are ask for the type 1 and type 2 errors to be controlled in the same way.
* Line 111: replace the bars "$||$" with a comma "," inside the TV operator.
* Line 136: I think you mean Proposition 3.1 (not Theorem 3.1).
* Line 139: I think you forgot the subscripts on the distributions inside the first sup.  Also, did you flip the order of the subscripts describing the asymptotics?

**Significance.**  In my opinion, the lower bounds are the most important contribution of the paper, and should be quite useful to the community at large.

**Quality.**  I think the paper is, overall, pretty solid.  The authors give some new lower bounds (which seem to be proven essentially by direct arguments, rather than information-theoretic ones).  They also give upper bounds that (the authors say outperform, in terms of testing risk, the state-of-the-art batch method of Diakonikolas et al. (2020), but more to the point ...) outperforms the state-of-the-art sequential algorithm of Daskalakis and Kawase (2017).

My only complaint, as far as quality goes, is that there aren't enough simulations.  I guess that is fine, for a theory paper.  But when words like "shows the preeminence of our sequential algorithm" (line 250) are being thrown around, then you really are claiming that your algorithm is better than all others, and so you had better have some pretty convincing numerical evidence backing that claim up.  Obviously, the current "experiment" (i.e., Figure 1) is not really thorough enough by any means to back up that claim ...

**Time Spent Reviewing:**

2.5

---

> ### Author Response · Authors · 2021-08-10
> **Comment to the review of Reviewer fx8n**
>
> Thank you for your very insightful review.
>
> We have cited the work of Howard et al which surveys the time uniform concentration inequalities and we think this is also the key for the paper suggested in the review. Nevertheless, we believe that this work merits to be cited and we will add a discussion about the difference between testing continuous distributions and discrete ones, and more generally about the different point of views on testing in CS and statistics.
> However, we believe that your criticism applies to most recent papers that our paper tries to follow and improve on, and does not disqualify our contributions. On the other hand, we focus on the TV distance in testing closeness problems because i) it characterizes the error probability when the tester sees a word consisting of i.i.d. samples from either the distribution $p$ or $q$ and wants to decide this distribution and ii) as pointed by the Reviewer fYML, testing in other distances is in general impossible. Finally,
> the optimal dependency in $\delta$ for testing closeness has not been known before the work of Diakonikolas et al. (2020), they used to invoke the amplification argument and thus multiply the complexity by $\log(1/\delta)$, for this reason we have compared our sequential algorithm to  the batch algorithm of Diakonikolas et al. (2020) and we believe the latter outperforms the previous algorithms.

---

> > ### Comment · Reviewer_fx8n · 2021-09-10
> > **thanks**
> >
> > Thanks a lot for your comments.  The information is good to see.  I'll leave my rating as-is.

---

### Decision · Program_Chairs · 2021-09-27

**Decision:**

Accept (Spotlight)

**Comment:**

The paper provides new upper and lower bounds of the sample complexity of testing closeness of distributions, which together highlight the advantages of a sequential approach to the problem. Despite some initial concerns over constant-factor separations, the reviewers have reached a consensus that the results of the paper are novel and significant, providing crisp complexity bounds on a fundamental learning-related problem.

In revising the manuscript, the authors should pay close attention to the reviewer’s feedback, particularly in regards to discussion of related literature.